# Do young dating app users and non-users differ in mating orientations?

**Juan Ramón Barrada**[1], **Ángel Castro**[1]*, **Elena Fernández del Río**[2], **Pedro J. Ramos-Villagrasa**[2]

**1** Faculty of Social and Human Sciences, Department of Psychology and Sociology, University of Zaragoza, Teruel, Spain, **2** Faculty of Labour and Social Sciences, Department of Psychology and Sociology, University of Zaragoza, Zaragoza, Spain

☯ These authors contributed equally to this work.
* castroa@unizar.es

## Abstract

In recent years, dating apps have changed the way people meet and communicate with potential romantic and/or sexual partners. There exists a stereotype considering that these apps are used only for casual sex, so those apps would not be an adequate resource to find a long-term relationship. The objective of this study was to analyze possible individual differences in the mating orientations (short-term vs. long-term) between users and non-users of dating apps. Participants were 902 single students from a mid-size Spanish university, of both sexes (63% female, and 37% male), aged between 18 and 26 years ($M = 20.34$, $SD = 2.05$), who completed a battery of online questionnaires. It was found that, whereas dating apps users had a higher short-term mating orientation than non-users (more frequent behavior, higher desire, and more positive attitude), there were no differences in the long-term orientation as a function of use/non-use. Considering this, dating apps are a resource with a strong presence of people interested on hooking-up while, simultaneously, not a bad (nor good) option for finding long-term love.

**Data Availability Statement:** The open database and code files for these analyses are available at the Open Science Framework repository (https://osf.io/phy4n/).

## Introduction

In recent years, dating apps have become a very popular tool to initiate contact with potential romantic and/or sexual partners [1]. Geolocation applications (e.g., Tinder, Grindr) have made it especially easy to communicate and meet other users who are geographically close [2]. It is estimated that more than one hundred million people around the world regularly use these apps, which has made online dating one of the main ways to find a partner today, especially among young people [3].

It is widely believed that dating apps are used exclusively for casual sex [4]. However, research on this subject suggests otherwise. In the last years, a considerable amount of research [2–9] is developed, showing that people use these apps for a wide variety of reasons, and that seeking sex is not the main one at all. The reasons given above for sex vary in different studies, including relational (e.g., friendship, love), entertainment (e.g., curiosity, boredom), and intrapersonal issues (e.g., self-validation, ease of communication).

**Funding:** This work was supported by the Ministry of Science, Innovation and Universities, Government of Spain, under grant PGC2018-097086-A-I00; and by the Government of Aragon (Group S31_20D). Department of Innovation, Research and University and FEDER 2014-2020, "Building Europe from Aragón". The funders had no role in study design, data collection and analysis, decision to publish, or preparation of the manuscript.

**Competing interests:** The authors declared that no competing interests exist.

The motivations for using dating apps are determined by the users' individual characteristics [1]. Sociodemographic variables (i.e., sex, age, and sexual orientation) are those with a higher relationship with the use of apps [9,10]. Specifically, past literature highlighted that men [6,10], and members of sexual minorities [6,10,11], present higher prevalence rates for the use of dating apps. Based on age, the most studied group and in which higher rates of app use is older youth, who tend to show a wide variety of motives to use it, seeking both entertainment and casual sex or romantic partner [2,4,10].

With respect to relationship status, while some authors have found that a large proportion of people in a relationship are dating apps users [4,12,13], other studies have found that being in a relationship shows a negative and large association with current (last three months) use, but not associated with previous use [10]. Those discrepancies can be partially explained by the timeframe considered to mark participants as dating apps users. For instance, Lefebvre [4] explicitly indicated that with her data collection protocol current relationship status of the participants may or may not reflect their status when using Tinder. Orosz et al. [13] considered as users those who had used Tinder at least once in their lifetime, so, again, current relationship status was not the same as status when using dating apps.

Personality traits is also related with the use of dating apps, but its relevance is lower [10]. A variable that seems with an important relationship with dating apps is sociosexuality, understood as an individual´s disposition to engage in sexual relations without closeness or commitment [14–18]. Some studies, like those of Botnen et al. [8] and Grøntvedt et al. [19] have found that people with unrestricted sociosexuality (i.e., short-mating orientation) report to use these apps more.

Continuing with the influence of individual differences, the literature has paid particular attention to mating preferences and orientations. Mating is a lifelong process [20,21] with great implications for future life [22,23]. Traditionally, its relevance has been emphasized during emerging adulthood, when decisions are often made about relationships and offspring, events that have a considerable impact on peoples´ lives [20,21]. Mating orientation, the individuals' stated interest in committed relationships and/or in brief or uncommitted sexual relationships [24], has usually been measured through a single dimension with two opposite poles: short-term versus long-term [21]. Short-mating orientation is characterized by the search for casual sexual partners and relationships of low emotional commitment [21,24,25], and traditionally has been identified with unrestricted sociosexuality. Long-term mating orientation, on the other hand, is characterized by the desire for romantic relationships of commitment, with a strong emotional investment in the relationship and, generally, with sexual exclusivity [26].

This traditional view of mating orientation has been criticized by some authors, such as Jackson and Kirkpatrick [24], who claimed that short-term and long-term orientation are not two opposing poles in a single dimension, but two dimensions that, while negatively related, can be and should be differentiated. Thus, for example, it is possible to desire or be involved into a stable relationship and maintain multiple sexual relationships without commitment [27,28]. It is also possible to have no interest in any kind of relationship.

The conception of sociosexuality has also be refined. Different researchers have shown the appropriateness of abandoning the classic unidimensional stance of short-term orientations [18] and paying attention to a multidimensional perspective [15]. This more fine-grained approach includes sociosexual behavior (i.e., past sociosexual behavior), attitudes (i.e., positive appraisal about casual sex), and desire (i.e., sexual arousal with people with whom no committed romantic relationship exists).

However, it is still common that researchers continue to study mating strategies like opposing poles and sociosexuality from a unidimensional approach when they analyze demographic and psychological correlates. There is still some theoretical confusion in the use of some terms.

For instance, Penke [29] defined restricted sociosexuality as the "tendency to have sex exclusively in emotionally close and committed relationships" and unrestricted sociosexuality as the "tendency for sexual relationships with low commitment and investment" (p. 622). This conceptualization assumes that (a) restricted and unrestricted sociosexuality define a single dimension and (b) that restricted is equivalent to long-term mating orientation and unrestricted to short-term orientation. While we agree with the first assumption, we have justified that short- and long-term mating orientation are not the two extremes of a single dimension. While unrestricted sociosexuality can be understood as interchangeable with short-term orientation, restricted sociosexuality is not long-term, but lack of short-term orientation.

Mating orientations can also differ based on different sociodemographic characteristics. Previous literature has argued that men show a greater short-term orientation, while women prefer long-term relationships [20,21,26], both for evolutionary reasons and for the still prevailing sexual double standard. The evolutionary reasons refer to sexual differences: men want to have sex with as many women as possible, while women are selective, looking for the most suitable candidate to procreate [30]. Regarding the sexual double standard, it refers to the different assessment of a sexual behavior depending on whether it is performed by a man or a woman (e.g., hook-up) [31]. It has also been found that people go changing progressively their preferences when they grow up, involving in long-term relationships [22]. Regarding sexual orientation, individuals who are part of sexual minorities, especially men, are much more likely to have short-term relationships than heterosexual people [32], perhaps because they are looking for a partner for different reasons to the procreation [33].

For all the above reasons, it seems that young people: (1) use dating apps for a variety and complexity of motives that go beyond the mere pursuit of casual sex; and (2) do not merely follow an exclusive short- or long-term orientation, but instead, both models can coexist. This study aims to determine possible differences in the mating orientation between young users and non-users of dating apps. That is, if it is accepted that it is relatively common to seek sex without commitment through dating apps, is this medium a good or bad option to find long-term romantic relationships? A condition for being an effective option would be that dating apps users are long-term oriented or, at least, as long-term oriented as the non-users. Up to now, there is limited and indirect information regarding this.

Recently, it has been found that Tinder users have a higher likelihood of forming romantic relationship longitudinally, but that this increased likelihood can be explained by Tinder users' personality and substance use characteristics [34]. That previous dating apps use is not related to currently being single [10] can be interpreted as indicative that users are not relationship-avoidant people. The associations between apps use and mating orientations will be assessed controlling the effect of sociodemographic characteristics (gender, age, sexual orientation) and assessing short-term mating orientation (sociosexuality) from a tridimensional approach (behavior, attitudes, desire).

## Materials and methods

### Participants and procedure

This study was part of a larger project carried out in a Spanish university that aimed to explore several aspects of the sexuality of young students. The initial sample comprised 1,996 participants. Five inclusion criteria were used: (a) studying a university degree (76 participants excluded); (b) aged between 18 to 26 years (128 participants excluded); (c) labeling themselves as woman or man (13 participants excluded); (d) correctly answering a control question (41 participants excluded; see below); and (e) being single at the time of the study (803 participants

excluded). The four first criteria were the same as those used in previous research with equivalent samples [10,14,35].

We discarded the participants involved in a relationship for two reasons. First, because among people in a relationship, those who had used apps in the last three months were a very small minority (*n* = 33, 4.1%), so its limited sample size prevented any further analysis. Second, because we understood that, among dating apps users, the profiles and motives of using dating apps of those who were or were not in a relationship had to be very different [36,37].

After applying these criteria, the final sample comprised 902 single university students (63% women, 37% men), aged between 18 and 26 (*M* = 20.34, *SD* = 2.05). Of these participants, 68.2% described themselves as heterosexual, 22.6% as bisexual, 7.1% as homosexual, and 2.1% as other orientations. Due to the small sample sizes of the non-heterosexual participants, those participants were grouped into a sexual minority category (31.8%).

Data were collected through the Internet with Google Forms in December 2019. The link to the survey was distributed through the student e-mail lists of the authors' university. The survey remained open for 14 days. Participants provided informed consent after reading the description of the study, where the anonymity of their responses was clearly stated. This procedure was approved by the Ethics Review Board for Clinical Research of the region (PI18/058). The present sample is part of a larger data set used in a previous investigation [10]. However, the data used for this study do not match either the research questions, the variables used, or the subset of data used.

## Measures

**Sociodemographic and dating apps use questionnaire.** We asked participants about their gender (woman, men, other), age, and sexual orientation (heterosexual, homosexual, bisexual, other). We also asked whether participants had used any dating app (e.g., Tinder, Grindr) in the three months prior to participating in the study. We used a timeframe of three months as what we considered a compromise between two needs: To consider current users while still having a large enough sample size. With longer timeframes, the meaning of 'current use' is diluted. With a much stricter timeframe, the number of current users would not be enough for the intended analysis, while the meaning of 'current use' could be misleading (consider the case if you ask for use in the last 24 hours and a very active user without Internet connection in the previous day).

**Sociosexual Orientation Inventory-Revised (SOI-R [15]).** This instrument has nine items that assess sociosexual orientation/short-term orientation on the basis of three dimensions: Behavior (e.g., "In the last twelve months, with how many different partners have you had sexual intercourse without having an interest in a long-term committed relationship with this person?"; α = .94 –all reported alphas correspond to values obtained with the current sample–), Attitudes (e.g., "Sex without love is OK"; α = .81), and Desire (e.g., "How often do you have fantasies about having sex with someone with whom you do not have a committed romantic relationship?"; α = .79). These items are rated on a nine-point scale, ranging from 1 = *0* to 9 = *20 or more* in the Behavior factor; from 1 = *strongly disagree* to 9 = *strongly agree* in the Attitudes factor; and from 1 = *never* to 9 = *at least once a day* in the Desire factor. We used the Spanish validation [38] with a modification in the Behavior dimension. While in the original Spanish validation, no specific time frame is provided, in the present data collection, we specified a 12-month period.

**Long Term Mating Orientation Scale (LTMO [24]).** This instrument has seven items that assess long-term mating orientations with a single component (e.g., "I hope to have a romantic relationship that lasts the rest of my life"; α = .87). These items are rated on a seven-

point scale, ranging from 1 = *strongly disagree* to 7 = *strongly agree*. Details about the questionnaire translation into Spanish and item wording can be found in the S1 Appendix.

**Control question.** Embedded in the LMTO as its 8th item and in order to check whether the participants paid enough attention to the wording of the items, we introduced an item asking the participants to respond to it with *strongly disagree*.

## Data analysis

The analyses were performed with R 4.0.2. Firstly, we computed descriptives and correlations between the different variables. The correlations between dichotomous variables (gender, sexual orientation, having used apps) with age and the four mating orientation scores were transformed to Cohen's *d* to facilitate their interpretation.

Secondly, we computed linear regression models, with mating orientation scores as criteria variables and gender, sexual orientation, age, and having used apps as predictors. As the metric of the dependent variables is not easy to interpret, we standardized them before the regression. In these models, regression coefficients indicate the expected change in standard deviation units.

No missing data were present in our database. The open database and code files for these analyses are available at the Open Science Framework repository (https://osf.io/phy4n/).

## Results

The associations among the different variables, with the descriptives, can be seen in Table 1. As could be expected, those with higher long-term orientation showed lower short-term orientation, but those relations were small ($r = -.35$, 95% CI [−.41,−.30], for SOI-R Attitude; $r = -.13$, 95% CI [−.19,−.06], for both SOI-R Behavior and Desire).

Of the participants, 20.3% ($n = 183$) reported having used dating apps in the last three months. Concerning sociodemographic variables, those using dating apps tended to be older ($d = 0.30$, 95% CI [0.14, 0.46]), men ($r = .08$, 95% CI [.02, .15]) and non-heterosexual ($r = -.20$, 95% CI [−.26,−.14]).

**Table 1. Bivariate relations of the different variables and descriptive statistics.**

|  | 1 | 2 | 3 | 4 | 5 | 6 | 7 | 8 |
|---|---|---|---|---|---|---|---|---|
|  | | | Pearson *r* [95% CI] | | | | | |
| 1. SOI-R Behavior | | | | | | | | |
| 2. SOI-R Attitude | **.45 [.39,.50]** | | | | | | | |
| 3. SOI-R Desire | **.28 [.22,.34]** | **.45 [.40,.50]** | | | | | | |
| 4. LTMO | **−.13 [−.19,−.06]** | **−.35 [-.41, -.30]** | **−.13 [−.19,−.06]** | | | | | |
| 5. Age | **.19 [.13,.26]** | **.12 [.06,.19]** | **.16 [.10,.22]** | .02 [−.04,.09] | | | | |
|  | | | Cohen's *d* [95% CI] | | | | | |
| 6. Men | −0.10 [−0.24, 0.03] | −0.07 [−0.20, 0.07] | **0.35 [0.22, 0.49]** | **0.18 [0.04, 0.31]** | 0.05 [−0.08, 0.19] | Pearson *r* [95% CI] | | |
| 7. Heterosexual | **−0.23 [−0.38, −0.09]** | **−0.25 [−0.39, −0.11]** | **−0.15 [−0.29, −0.01]** | **0.16 [0.02, 0.30]** | 0.13 [−0.01, 0.27] | .01 [−.06,.07] | | |
| 8. Apps used | **0.83 [0.66, 1.00]** | **0.52 [0.35, 0.68]** | **0.50 [0.33, 0.66]** | −0.11 [−0.27, 0.06] | **0.30 [0.14, 0.46]** | **.08 [.02,.15]** | **−.20 [-.26, -.14]** | |
| Mean | 6.31 | 18.7 | 13.11 | 35.75 | 20.34 | .37 | .68 | .20 |
| Standard deviation | 4.64 | 6.65 | 5.49 | 9.20 | 2.05 | .48 | .47 | .40 |

Notes: SOI-R = Sociosexual Orientation Inventory-Revised; LTMO = Long Term Mating Orientation Scale; CI = confidence interval; Men = dummy variable where *women* = 0 and *men* = 1; Heterosexual = dummy variable where *sexual minority* = 0 and *heterosexual* = 1; Apps used = dummy variable indicating whether any dating app was used in the three months prior to participating in the study. Bold values correspond to statistically significant associations ($p < .05$)

**Table 2. Multiple regression analysis of the different mating orientation scales.**

|  | SOI–R Behavior | | | SOI–R Attitude | | | SOI–R Desire | | | LTMO | | |
|---|---|---|---|---|---|---|---|---|---|---|---|---|
|  | $R^2_{adj}$ | $F$ | $p$ | $R^2_{adj}$ | $F$ | $p$ | $R^2_{adj}$ | $F$ | $p$ | $R^2_{adj}$ | $F$ | $p$ |
|  | 0.13 | 34.78 | < .001 | 0.06 | 14.58 | < .001 | 0.08 | 20.05 | < .001 | 0.01 | 3.37 | .009 |
|  | $b$ [95% CI] | SE | $p$ | $b$ [95% CI] | SE | $p$ | $b$ [95% CI] | SE | $p$ | $b$ [95% CI] | SE | $p$ |
| Intercept | **−1.62 [−2.23, −1.01]** | **0.31** | **< .001** | **−0.98 [−1.62, -0.34]** | **0.32** | **.003** | **−1.51 [−2.14, −0.88]** | **0.32** | **< .001** | −0.37 [−1.03, 0.28] | 0.33 | .262 |
| Men | **−0.16 [−0.28, −0.03]** | **0.06** | **.015** | −0.10 [−0.23, 0.03] | 0.07 | .132 | **0.31 [0.18, 0.45]** | **0.07** | **< .001** | **0.18 [0.05, 0.32]** | **0.07** | **.008** |
| Hetero | −0.13 [−0.26, 0.01] | 0.07 | .062 | **−0.19 [-0.33, -0.05]** | **0.07** | **.008** | −0.10 [−0.24, 0.04] | 0.07 | .144 | 0.14 [0.00, 0.28] | 0.07 | .054 |
| Age | **0.08 [0.05, 0.11]** | **0.02** | **< .001** | **0.05 [0.02, 0.08]** | **0.02** | **.001** | **0.07 [0.04, 0.10]** | **0.02** | **< .001** | 0.01 [−0.02, 0.04] | 0.02 | .490 |
| Apps used | **0.73 [0.57, 0.88]** | **0.08** | **< .001** | **0.44 [0.28, 0.60]** | **0.08** | **< .001** | **0.39 [0.23, 0.55]** | **0.08** | **< .001** | −0.10 [−0.26, 0.07] | 0.09 | .251 |

Notes: SOI-R = Sociosexual Orientation Inventory-Revised; LTMO = Long Term Mating Orientation Scale; CI = confidence interval; Men = dummy variable where *women* = 0 and *men* = 1; Heterosexual = dummy variable where *sexual minority* = 0 and *heterosexual* = 1; Apps used = dummy variable indicating whether any dating app was used in the three months prior to participating in the study. Bold values correspond to statistically significant coefficients ($p < .05$).

With respect to mating orientation, those using apps showed higher scores in all three SOI-R dimensions, mainly in short-term behavior (*d*s in the range [0.50, 0.83]). All previously reported associations were statistically significant (*p*s < .001). Importantly, no statistically significant differences in long-term orientation scores were found as a function of using or non-using dating apps and the confidence interval only included what could be considered as null or small effect sizes (*d* = −0.11, 95% CI [−0.27, 0.06], *p* = .202).

While men presented a higher sociosexual desire than women (*d* = 0.35, 95% CI [0.22, 0.49], *p* < .001) and higher long-term orientation scores (*d* = 0.18, 95% CI [0.04, 0.31], *p* = .010), no statistically significant difference was found in short-term behavior (*d* = −0.10, 95% CI [−0.24, 0.03], p = .146) or attitude (*d* = −0.07, 95% CI [−0.20, 0.07], *p* = .333). Sexual minority participants presented higher scores than heterosexual participants in all three dimensions of short-term orientation (behavior: *d* = 0.23, 95% CI [0.09, 0.38], *p* = .001; attitude: *d* = 0.25, 95% CI [0.11, 0.39], *p* < .001; desire: *d* = 0.15, 95% CI [0.01, 0.29], *p* = .035), while heterosexual participants showed a higher long-term orientation (*d* = 0.16, 95% CI [0.02, 0.30], *p* = .023). Older participants showed higher short-term orientation scores (behavior: *r* = .19, 95% CI [.13,.26]; attitude: *r* = .12, 95% CI [.06,.19]; desire: *r* = .16, 95% CI [.10,.22]; all *p*s < .001), but age was not related to long-term orientation (*r* = .02, 95% CI [−.04,.09], *p* = .462).

Results of the four regression models are shown in Table 2. While controlling for gender, sexual orientation, and age, the pattern of results for dating apps use remained basically unchanged with respect to bivariate associations. Given the goals of our manuscript, we will focus our attention on the differences between users and non-users of dating apps. Those using apps, with respect those not using them, showed a score 0.73 standard deviations higher in short-term Behavior (95% CI [0.57, 0.88], $R^2_{adj} = .13$), 0.44 standard deviations higher in short-term Attitude (95% CI [0.28, 0.60], $R^2_{adj} = .06$), and 0.39 standard deviations higher in short-term Desire (95% CI [0.23, 0.55], $R^2_{adj} = .08$; all *p*s < .001). However, no significant effect was present for apps use in long-term orientation (*b* = −0.10, 95% CI [−0.26, 0.07], *p* = .251).

## Discussion and conclusions

The development of dating apps in recent years has generated some debates, especially related to the motivations for their use. Usually, it has been considered that dating apps were used for casual sex, although other studies have shown that the reasons for their use are more diverse

and complex and may include, among others, the search for long-term romantic relationships [2–9]. In the attempt to contribute information to this debate, the objective of this study was to analyze possible differences in the mating orientations in a sample of single young university students depending on whether or not they were users of dating apps.

In response to the main objective of the study, differences were found between users and non-users of dating apps in the three dimensions of short-term orientation–especially in socio-sexual behavior–but not in long-term orientation. That is, among app users, it is comparatively easier to find more unrestricted sexually-oriented people, whereas users and non-users do not differ in their interest in maintaining a long-term romantic relationship.

This allows several conclusions to be drawn. First, according to the existing literature and the constructs evaluated, it seems logical that those who use dating apps, many who are open to casual sex, will score higher in the three dimensions of sociosexuality than those who do not use them [9,17]. Secondly, the absence of differences in the long-term orientation indicates that the orientations are not exclusive and contrary to each other [24,25]. Dating apps users, although open to short-term relationships, are not reluctant to long-term mating. This con-verges with previous results as longitudinal higher likelihood of forming romantic the longitu-dinal by Tinder users [34] or that previous use is not related to being single [10]. This pattern of results opens the door to the perception that there may be flexibility in mating orientations and preferences and that they can coexist simultaneously in people seeking both a casual rela-tionship and a romantic relationship [24].

Thirdly, among the contributions of the article should be highlighted the assessment of sociosexuality from a multidimensional point of view, distinguishing between behavior, atti-tudes, and desire, following the recommendations of other authors [15,38]. It has been shown that the three dimensions of the construct, understood as short-term orientation, correlate positively and directly with each other and inversely with the long-term orientation, although the intensity of the association varies, being more powerful in attitudes and less powerful in sociosexual behavior and desire. This points to the need to step away from the conceptualiza-tion of unrestricted sociosexuality as equal to short-term mating orientation and restricted sociosexuality as equal to long-term mating orientation [29]. As we previously noted, restricted sociosexuality is better understood as lack of short-term orientation, what is not equivalent to long-term orientation.

In addition, as regards the prevalence of use of dating apps among the participants in the last three months, 20.3% of users were found among those who were singles (12.7% of the total sample), which represents a medium-low prevalence compared to other studies [2,3,5–7], although it should be noted that, in these studies, sampling was aimed at finding people who used dating apps [1].

Of the other results obtained, the most relevant, although it was beyond the main objective of the study, were the differences found in the long-term orientation between single men and women. Contrary to our expectations, men scored slightly higher than women in this variable. A greater long-term orientation had usually been found in women [16,20,21,24,28]. As this is the first study of its kind to be carried out in Spain, it is difficult to identify the causes and determine whether this is a cultural pattern or whether it simply responds to the characteristics of the study sample. In any case, this result seems to suggest that women are increasingly own-ers of their sexuality and of the decisions that have to do with it, moving away from the effects of traditional double standard [23].

Also contrary to expectations, a relationship was found between age and short-term orien-tation, but not with long-term relationships. The existing literature defends that people go changing progressively their preferences when they grow up, involving in long-term relation-ships [22]. However, due to the limited age range of the participants of the present study, this

variation cannot be seen in the interests and behaviors of university students. Finally, we found that while heterosexual participants were more oriented to long-term mating, sexual minorities were more inclined towards short-term mating. This result was already present in the literature [33].

The study has a number of limitations. The use of dating apps was evaluated without delving into the variety of uses, from those who used it on a single afternoon as a joke among friends to those who used it for months looking for a romantic relationship. So, what we treated a unitary (self-reported) behavior–dating apps use–included, in fact, important differences in motivations or intensity. Other limitations were related to the representativeness of the sample and the generalization of the results. Among the final participants, the sample was mostly female, aged between 18 and 26, single and from a single university, making the results difficult to generalize to all university students and, still less to young non-university students.

Concerning to sexual orientation, two aspects should be noted. First, the high proportion of participants from sexual minorities, more than 30% of the final sample. This could be considered as a lack of representativeness of our sample. We consider that an alternative interpretation is possible. This study shares with previous studies the same sampling approach and population (Spanish university students with the same age range and from the same university). We will show the time of data collection and the proportion of sexual minority participants: November 2018, 27.0% [14], December 2017, 22.5% [9], May 2016, 14.7% [38], April 2016, 12.7% [35], October 2013, 8.6% [39]. A clear trend is found. The proportion of sexual minority participants is steadily increasing in our samples.

We can imagine two options to explain this. First, our surveys are not just biased by sexual orientation (higher probability of participation for non-heterosexual people), but also that bias is growing. We cannot find any theoretically plausible explanation for this potential change of bias across time. Second, in fact in the population of university students (Spain, a single university) the presence of non-heterosexuality is increasing. This second alternative would imply that the large number of non-heterosexual participants is not a problem of representativeness of the samples.

This hypothesis may be supported by data on the prevalence of persons from sexual minorities found in other studies, which can be exemplified in that of Rahman et al. [40], who assessed the prevalence of women´s and men´s sexual orientation in 28 nations and found similar proportions to those of the present study, both in Spain (73% vs. 27%) and in other countries (e.g., United States, Australia, Finland). There seems to be a trend toward greater self-identification as a member of sexual minorities, paralleling the decrease in stigma and the improvement in the quality of life of these people, especially in countries with more tolerant laws, as is the case in Spain [41]. However, further research is needed to clarify this point. And, in any case, in our regression analyses we included sexual orientation as covariate. In addition, to facilitate the analyses, we decided to group participants into heterosexuals and non-heterosexuals, thus losing the nuances related to the behavior of members of sexual minorities.

Similarly, our study shares with other studies based on self-selected samples and self-reported measures the fact that the results may be limited by response and recall bias. Finally, like most literature on the subject, this study is cross-sectional. It would be interesting to design longitudinal investigations, to assess the development and stability/change, both in the use of dating apps and in mating orientations and their associations.

Despite these limitations, the study is considered to meet the objective posed and answers the question that prompted it. Users of dating apps have a greater short-term orientation than non-users, with no differences in long-term orientation. Thus, it can be said that both types of orientations and relationships are expressions of sexuality that can coexist, that they are not considered as excluding and that, regardless of the type of people's sexual relations, the

important thing is that they are healthy, performed in a context of mutual respect. With regard to the objective of the study, summarizing: dating apps seem to be good for casual sex and not bad for finding long-term love.

## Supporting information

**S1 Appendix.**
(PDF)

## Author Contributions

**Conceptualization:** Juan Ramón Barrada, Ángel Castro, Elena Fernández del Río, Pedro J. Ramos-Villagrasa.

**Data curation:** Juan Ramón Barrada.

**Formal analysis:** Juan Ramón Barrada.

**Funding acquisition:** Juan Ramón Barrada, Ángel Castro.

**Investigation:** Ángel Castro.

**Methodology:** Juan Ramón Barrada.

**Project administration:** Ángel Castro.

**Resources:** Ángel Castro.

**Supervision:** Elena Fernández del Río, Pedro J. Ramos-Villagrasa.

**Validation:** Juan Ramón Barrada.

**Visualization:** Elena Fernández del Río, Pedro J. Ramos-Villagrasa.

**Writing – original draft:** Juan Ramón Barrada, Ángel Castro, Elena Fernández del Río, Pedro J. Ramos-Villagrasa.

**Writing – review & editing:** Juan Ramón Barrada, Ángel Castro, Elena Fernández del Río, Pedro J. Ramos-Villagrasa.

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
