## [Decision Letter · Decision Letter 0]

26 Oct 2020

PONE-D-20-29867

Dating apps: Good for hooking-up, not bad for finding long-term love

PLOS ONE

Dear Dr. Castro,

Thank you for submitting your manuscript to PLOS ONE. After careful consideration, we feel that it has merit but does not fully meet PLOS ONE’s publication criteria as it currently stands. Therefore, we invite you to submit a revised version of the manuscript that addresses the points raised during the review process. The reviewers point our several methodological, sampling, analytical, and theoretical concerns. For example, Reviewer 1 points out that your sample has a particularly high number of non-heterosexuals which may (1) influence the representativeness of the sample but (2) also encourage more detailed analyses. Reviewer 2 points out conceptual issues like treating having downloaded the app as a behavior when it is behavioroid at best (i.e., self-reported behaviors). Despite these shortcomings, I am confident you can improve your paper from their comments and thus I rendered this decision.

We look forward to receiving your revised manuscript.

Kind regards,

Peter Karl Jonason

Academic Editor

PLOS ONE

Journal Requirements:

2. Please include additional information regarding the survey or questionnaire used in the study and ensure that you have provided sufficient details that others could replicate the analyses.

For instance, if you developed a questionnaire as part of this study and it is not under a copyright more restrictive than CC-BY, please include a copy, in both the original language and English, as Supporting Information.

Reviewers' comments:

Reviewer's Responses to Questions

**Comments to the Author**

1. Is the manuscript technically sound, and do the data support the conclusions?

Reviewer #1: Yes

Reviewer #2: Yes

2. Has the statistical analysis been performed appropriately and rigorously? 

Reviewer #1: No

Reviewer #2: No

3. Have the authors made all data underlying the findings in their manuscript fully available?

Reviewer #1: Yes

Reviewer #2: Yes

4. Is the manuscript presented in an intelligible fashion and written in standard English?

Reviewer #1: Yes

Reviewer #2: Yes

5. Review Comments to the Author

Reviewer #1: Title: Dating apps: Good for hooking-up, not bad for finding long-term love

Using a large sample of single Spanish university students (n = 902) this paper investigates how individual differences in mating orientation differs between those using dating apps and non-users. My reading the of the introduction is that they want to examine the relationship between mating short-term and long-term orientations and the use of dating apps. Instead of looking at factors that may affect the use of dating apps as studied in the prior literature, they predict preferences for long-term and short-term mating orientation from the use of dating apps. Specifically, they regress age gender, sexual orientation, and the use of dating apps on SOI-R, SOI-Behavior, SOI-Attitudes, and SOI-Desire, and Long-term Mating Orientation (LTMO). E.g., in the abstract they claim that there were no differences in the long-term orientation as a function of use/non-use. I find this problematic from a theoretical point, because they seem to assume that dating app use may affect mating orientation rather than the other way around. Mating orientation as measured with SOI-R and MSOI are measures of preferences and personality characteristics. It is very unlikely that such preferences are a result of the use of dating apps.

The method section gives a good impression of the sample characteristics. The sample of students seem to diverge notably from a general student population with regard to the proportion of non-heterosexual students. In the study almost 1/3 of the students were categorized as being sexual minority. This is not discussed in the study limitations (lack of representativeness). There is also reason to believe that heterosexuals and sexual minorities differ with regard to their various short term and long-term mating strategies, possibly moderated by gender (sex x sexual orientation interaction).

There are to date not many studies on mobile app use. Unfortunately, the authors have missed several of them. There are some studies on the motives of use (Botnen et al., 2018; LeFebvre, 2018; Timmermans et al., 2017) and there is one study on the effects of app use on having casual sex (Grøntvedt et al., 2019) and on likelihood of getting into long-term relationships (Erevik et al., 2020). A very central finding is that sociosexual orientation (the three SOI-R components) is found to predict current use of dating apps over and above the effects of other relevant factors (Botnen et al., 2018). Where relevant, several of the above studies need to be included in the introduction part of this study. I have listed the references below.

Given the solidity of the measurements used for this study, and the high number of sexual minority students, the authors should be able to re-analyze and present their findings more in line with the prior studies and theoretical models with particular focus on possible differences between heterosexual and sexual minority students of both gender. However, as the paper reads now, I would not recommend it for publication in PLOS ONE.

Minor:

In the description of inclusion criteria, I would prefer a more straightforward presentation (e.g., use ‘single’, rather than ‘not having a partner at the time of the study’).

Relevant references:

Botnen, E. O., Bendixen, M., Grøntvedt, T. V., & Kennair, L. E. O. (2018). Individual differences in sociosexuality predict picture-based mobile dating app use. Personality and Individual Differences, 131, 67-73. doi:https://doi.org/10.1016/j.paid.2018.04.021

Erevik, E. K., Kristensen, J. H., Torsheim, T., Vedaa, Ø., & Pallesen, S. (2020). Tinder Use and Romantic Relationship Formations: A Large-Scale Longitudinal Study. Frontiers in Psychology, 11(1757). doi:10.3389/fpsyg.2020.01757

Grøntvedt, T. V., Bendixen, M., Botnen, E. O., & Kennair, L. E. O. (2019). Hook, Line and Sinker: Do Tinder Matches and Meet Ups Lead to One-Night Stands? Evolutionary Psychological Science. doi:10.1007/s40806-019-00222-z

LeFebvre, L. E. (2018). Swiping me off my feet: Explicating relationship initiation on Tinder. Journal of Social and Personal Relationships, 35(9), 1205-1229. doi:10.1177/0265407517706419

Reviewer #2: The topic of the article is of importance for potential as well as actual dating app users, and is of everyday interest. The article tries to distinguish between recent users and non-users regarding the attitude towards short-term mating and long-term mating of a Spanish sample of (single) students.

Some general aspects:

Behavior is not equal to asking if someone had the app installed during the last 3 months. Differences, like you mentioned out, might

Introduction:

49-53:

Mating is not limited to (older) youth, further Ranzini and Lutz had a age range from 16 to 40, LeFebvre from 18 to 34 years.

Dating orientation is of interest (DAs/tinder) do not differ. You can look for both.

55-56ff:

Mating is not limited to youth, esp. not compared to the evolutionary background mentioned by Buss. Following Buss’ idea, which is hard to falsify, maybe, the ongoing concept of looking at short-term and long-term mating as not being opposite poles (as e.g. Kirkpatrick [14]) could already be mentioned in this context or refer to this statement. Further, potential argumentation based on evolutionary psychology often overlooks cultural and social component, explaining the majority of some effects (Eagly & Wood, 1999, Norenzayan & Heine, 2005). Also true for 81-84 [resp. 24 & 25]. Problematic on this is also your big part of non-heterosexuals (maybe more using for motive, mentioned by your literature, and not by evolutionary reasons).

(98-100: Using “i.e.” leaves the reader wondering why used three times in a row. Leaving out?)

94-96 & 178-181-:

“A condition for being an effective option would be that dating apps users are long-term oriented or, at least, as long-term oriented as the general population.” This question can neither be addressed, nor answered in the sample, see next point.

Materials & Methods

Participants & Procedure

A major point is the claiming for generalizability of the sample while there were two exclusions made: Once: Age was limited from 18 to 26. Second: Only people considering themselves as men or women were included in the analyses (they also could be dropped and considered as missing in the regression).

Plus: The considerations seems to be hetero- and non-heterosexual (maybe using the Kinsey Scale next time?), as roughly one out of four of the sample was considering themselves as not completely heterosexual and no other preferences were offered, this label would seem to be more fitting and also look less judging, see also 276-277, were this wording was used.

This means: The very limited category of young, studying, hetero- or non-heterosexual men and women is not easy to be generalized beyond itself and it cannot answer the questions about the long-term orientation differences of the population and not answer it in itself. E.g. the preference for older mates is not lacking the older part in the sample. Next: The limited age span is rather a categorical than a metric variable and therefore overemphasizing potential effects of age.

People in a relationship are usually typical dating app users (Freyth & Batinic, 2021; Hobbs et al, 2017; LeFebvr, 2018; Orosz et al. 2016), regards Grindr this can be assumed to, as more relationships are considered to be open. Freyth & Batinic further could not find a difference of the relationship status regarding using and not-using dating apps, but also no difference in actual dating app using behavior.

Concluding: The assumptions for excluding the data seems arbitrary and partwise odd. The analysis would be easier to generalize if the sample wasn’t reduced this way or theoretical reasons to do so would be offered.

Measures

The question about using apps in the last 3 months is probably a too short window and giving no information about the way of using the apps. Probably it is useful to talk about “recent users”?

Data Analysis/Results

176-181 Users/non-users on long-term mating orientation: “considered as small effect sizes”. As the CI includes zero, no further reports would be necessary. Further, this part of reporting could be headlined separately (descriptive?), before the regression is presented.

No conclusions should be drawn in the results, e.g. 184: “short-term behavior”. The analysis is referring to the SOI-R, which is considered to be a short-term mating measure, yet the results should be referring to the scale.

Also, the whole section could be shortened, as just the correlation table is reported from 168-193.

On the Regression: First, it looks odd compared to Castro (2020), that the analysis was not included in there, and/or second, that is was compared to the results. As Castro did show, differences regarding age, gender and sexual minority/heterosexuality have already be shown in the data set.

Next, the SOI-R should be included in one single multilinear regression, including long-term mating orientation. As mentioned before, age should probably be controlled in the model (once excluded and then once included, but maybe as dichotomous variable by e.g. mediansplit).

The SOI-R did report differences between men and women (expected). Yet in the presented regression (table 2) the results are unexpected towards Penke & Asendorpf (2008, p. 14). How come? How does this look in the complete model?

Discussion:

e.g. 222-229: The discussion and also the theory are referring to the motives of users. Yet no motives are analyzed, but attitudes. The link between both is missing in the article, please provide some further literature or concentrate on attitudes.

Severe: The reference of single university students is missing before (title or at least in the first half of the abstract).

235: “Conclusions”. As correlations are presented and hypothesis tested in a cross-sectional study, the wording might be chosen more thoughtful. And: The sample was heavily reduced compared to the presented references.

250-254: Better part of the sample section. The differences to other samples should then be mentioned in the limitations.

255-263: Might it be possible, as pointed out before (e.g. Ranzini) that male users misrepresent themselves in the apps and also in the questionnaire as a tactique? See Freyth & Batinic (2021).

263ff: Age correlations might be arbitrary (see above).

94-96, 290: One might wonder why the motivation and the attitude are sufficient to report successful behavior on finding long-term partners, as looking, searching and finding are different behavioral styles.

If further variable are available in the data, please include in the analyses and results.

Mentioned sources:

Eagly, A. H., & Wood, W. (1999). The origins of sex differences in human behavior: Evolved dispositions versus social roles. American psychologist, 54(6), 408.

Freyth, L., & Batinic, B. (2021). How bright and dark personality traits predict dating app behavior. Personality and Individual Differences, 168, 110316. doi.org/10.1016/j.paid.2020.110316

Norenzayan, A., & Heine, S. J. (2005). Psychological universals: What are they and how can we know?. Psychological bulletin, 131(5), 763.

6. PLOS authors have the option to publish the peer review history of their article (what does this mean?). If published, this will include your full peer review and any attached files.

Reviewer #1: No

Reviewer #2: **Yes: **Lennart Freyth

---

## [Author Response · Author response to Decision Letter 0]

4 Dec 2020

PONE-D-20-29867

Dating apps: Good for hooking-up, not bad for finding long-term love

PLOS ONE

First of all, we would like to thank the editor and the reviewers for the time and effort invested in reviewing this manuscript. Your comments have been very useful to improve our work. 

COMMENTS BY REVIEWER #1

Using a large sample of single Spanish university students (n = 902) this paper investigates how individual differences in mating orientation differs between those using dating apps and non-users. My reading of the introduction is that they want to examine the relationship between mating short-term and long-term orientations and the use of dating apps. Instead of looking at factors that may affect the use of dating apps as studied in the prior literature, they predict preferences for long-term and short-term mating orientation from the use of dating apps. Specifically, they regress age gender, sexual orientation, and the use of dating apps on SOI-R, SOI-Behavior, SOI-Attitudes, and SOI-Desire, and Long-term Mating Orientation (LTMO). E.g., in the abstract they claim that there were no differences in the long-term orientation as a function of use/non-use. I find this problematic from a theoretical point, because they seem to assume that dating app use may affect mating orientation rather than the other way around. Mating orientation as measured with SOI-R and MSOI are measures of preferences and personality characteristics. It is very unlikely that such preferences are a result of the use of dating apps. 

Response: We would like to thank Reviewer 1 for the time and effort invested in reviewing our manuscript and for his/her comments. Your suggestions have been very useful to improve our work. 

Thank you for the opportunity to clarify this point. Our research objective is to verify if there are differences between users and non-users of dating apps in mating orientations (short-term and long-term). We have tried to make this clearer by changing the title of our manuscript, which is now "Do young dating app users and non-users differ in mating orientations?". In our analysis, no specific causality (personality � apps use or apps use � personality) is intended. With a cross-sectional design, we cannot establish any. From our point of view, we are NOT assuming that dating app use may affect mating orientation. We have used the analysis that we consider that is better suited to response to our research question. In the Abstract (p. 2, lines 30-33), for instance, we wrote:

It was found that, whereas dating apps users had a higher short-term mating orientation than non-users (more frequent behavior, higher desire, and more positive attitude), there were no differences in the long-term orientation as a function of use/non-use.

Here, as well as all through the manuscript, we have presented the results as descriptive, not with any specific assumption of causes and consequences.

The method section gives a good impression of the sample characteristics. The sample of students seem to diverge notably from a general student population with regard to the proportion of non-heterosexual students. In the study almost 1/3 of the students were categorized as being sexual minority. This is not discussed in the study limitations (lack of representativeness).

Response: Thank you very much for your comments and suggestion. Our sample is indeed different from that of studies carried out in other geographical and cultural contexts, with a higher proportion of participants from sexual minorities. However, we are unsure about why this should be indicative of lack of representativeness, although, for sure, we also cannot guarantee representativeness of our sample. The sampling for this study is equivalent or very similar to the sampling from previous studies from the same research group (see Table below).

Data Collection Proportion of non-heterosexual Publication

December 2019 31.8% Present manuscript

November 2018 27.0% Fernández del Río et al. (2019) - Personality and short-term mating

December 2017 22.5% Barrada & Castro (2020) - Tinder users

May 2016 14.7% Barrada et al. (2018) - SOI-R validation

April 2016 12.7% Barrada et al. (2019) - Online sexual activities

October 2013 8.6% Castro & Santos-Iglesias (2016) - Sexual behavior and risks

A clear trend is found. The proportion of non-heterosexual participants is increasing in our samples. We can imagine two options to explain this. First, our surveys are not just biased by sexual orientation (higher probability of participation for non-heterosexual people), but also that bias is growing. We cannot find any theoretically plausible explanation for this. Second, in the population of university students (Spain, a single university), the presence of non-heterosexuality is increasing. Further research is needed to clarify this point.

We have introduced this in the Discussion section (pp. 15-16, lines 341-360):

Concerning to sexual orientation, two aspects should be noted. First, the high proportion of participants from sexual minorities, more than 30% of the final sample. This could be considered as a lack of representativeness of our sample. We consider that an alternative interpretation is possible. This study shares with previous studies the same sampling approach and population (Spanish university students with the same age range and from the same university). We will show the time of data collection and the proportion of sexual minority participants: November 2018, 27.0% [14], December 2017, 22.5% [10], May 2016, 14.7% [37], April 2016, 12.7% [34], October 2013, 8.6% [38]. A clear trend is found. The proportion of sexual minority participants is steadily increasing in our samples. We can imagine two options to explain this. First, our surveys are not just biased by sexual orientation (higher probability of participation for non-heterosexual people), but also that bias is growing. We cannot find any theoretically plausible explanation for this potential change of bias across time. Second, in fact in the population of university students (Spain, a single university) the presence of non-heterosexuality is increasing. This second alternative would imply that the problem of representativeness is more apparent that real. Further research is needed to clarify this point. In any case, in our regression analyses we included sexual orientation as covariate. In addition, to facilitate the analyses, we decided to group participants into heterosexuals and non-heterosexuals, thus losing the nuances related to the behavior of members of sexual minorities. 

There is also reason to believe that heterosexuals and sexual minorities differ with regard to their various short term and long-term mating strategies, possibly moderated by gender (sex x sexual orientation interaction).

Response: Concerning the initial part of this comment ("There is also reason to believe that heterosexuals and sexual minorities differ with regard to their various short term and long-term mating strategies") we fully agree. That is the reason why those variables were already included in the multiple regression analyses.

We also agree with the idea that sexual orientation and gender may interact. In the next tables we show the results of the linear regression analysis when these interactions are included in the models.

 SOI–R Behavior SOI–R Attitude SOI–R Desire LTMO

 Without Interaction Term Men × Hetero

 R_adj^2 F p R_adj^2 F p R_adj^2 F p R_adj^2 F p

 0.13 34.78 <.001 0.06 14.58 <.001 0.08 20.05 <.001 0.01 3.37 .009

 b SE p b SE p b SE p b SE p

Intercept –1.62 0.31 <.001 –0.98 0.32 .003 –1.51 0.32 <.001 –0.37 0.33 .262

Men –0.16 0.06 .015 –0.10 0.07 .132 0.31 0.07 <.001 0.18 0.07 .008

Hetero –0.13 0.07 .062 –0.19 0.07 .008 –0.10 0.07 .144 0.14 0.07 .054

Age 0.08 0.02 <.001 0.05 0.02 .001 0.07 0.02 <.001 0.01 0.02 .49

Apps used 0.73 0.08 <.001 0.44 0.08 <.001 0.39 0.08 <.001 –0.10 0.09 .251

 With Interaction Term Men × Hetero

 R_adj^2 F p R_adj^2 F p R_adj^2 F p R_adj^2 F p

 0.13 29.09 <.001 0.06 12.32 <.001 0.08 16.37 <.001 0.02 3.89 .002

 b SE p b SE p b SE p b SE p

Intercept –1.71 0.31 <.001 –0.91 0.33 .005 –1.46 0.32 <.001 –0.47 0.33 .1611

Men 0.07 0.12 .550 –0.28 0.12 .021 0.19 0.12 .111 0.43 0.12 <.001

Hetero –0.01 0.08 .899 –0.28 0.09 .001 –0.17 0.09 .053 0.27 0.09 .003

Men × Hetero –0.33 0.14 .018 0.26 0.14 .075 0.18 0.14 .203 –0.36 0.15 .015

Age 0.08 0.02 <.001 0.05 0.02 .001 0.07 0.02 <.001 0.01 0.02 .466

Apps used 0.70 0.08 <.001 0.46 0.08 <.001 0.40 0.08 <.001 –0.12 0.09 .157

 SOI–R Behavior SOI–R Attitude SOI–R Desire LTMO

 Without Interaction Term Men × Hetero

 R_adj^2 F p R_adj^2 F p R_adj^2 F p R_adj^2 F p

 0.13 34.78 <.001 0.06 14.58 <.001 0.08 20.05 <.001 0.01 3.37 .009

 b SE p b SE p b SE p b SE p

Apps used 0.73 0.08 <.001 0.44 0.08 <.001 0.39 0.08 <.001 –0.10 0.09 .251

 With Interaction Term Men × Hetero

 R_adj^2 F p R_adj^2 F p R_adj^2 F p R_adj^2 F p

 0.13 29.09 <.001 0.06 12.32 <.001 0.08 16.37 <.001 0.02 3.89 .002

 b SE p b SE p b SE p b SE p

Apps used 0.70 0.08 <.001 0.46 0.08 <.001 0.40 0.08 <.001 –0.12 0.09 .157

In the upper table the full model is shown. In the lower table, just the summary statistics of the model and the information about apps use are offered. The interaction term Men × Hetero was statistically significant in two out of four cases (SOI-R Behavior, b = –0.33, p = 0.18; LTMPO, b = –0.36, p = 0.015), while not in for the other two criteria variables (SOI-R Attitude, b = 0.26, p = .075; SOI-R Desire, b = 0.18, p = .203). Importantly, the differences in the coefficients of dating apps for the models with or without that interaction term were negligible, never larger than 0.03, and the pattern of statistically significance was unaffected.

We have kept the models without that interaction term for the following reasons:

- Our focus is on the differences among users/non-users. Those differences are unaffected by the inclusion of that interaction.

- From our point of view, the remaining coefficients of the model should not be interpreted, so it is not relevant whether they have been changed. While the differences between dating apps users and non-users, controlling for gender, sexual orientation, and age are theoretically interpretable (and those are where we can find the response to our research question), it is much harder to interpret the differences, for instance, between women and men controlling for dating apps use and the rest of variables. This latter coefficient does not represent differences among genders, but differences in very specific conditions, while holding constant dating apps use. That comparison, probably, is not the key comparison that would be relevant for readers.

- We could not provide a theoretical reason to only include the interaction term Men × Hetero while not all the other potential interactions (Men × Age, Men × Apps used...). Including all those interaction terms would reduce the statistical power for the coefficient that is the only one that we want to interpret, the one for Apps used.

- Previous research have found that interaction terms are associated with lower replicability. See, for instance, Altmejd et al. (2019), or Open Science Collaboration (2015).

We must stress that the goal of our manuscript was not to predict mating orientation, but whether there are differences in mating orientations when comparing dating apps users and non-users. The proposed analyses are very interesting, but for addressing a research question that is not ours. We expect that we have been able to provide evidence that the inclusion or exclusion of the proposed interaction leads to no change concerning our goals –so nothing is lost by not incorporating it–, but including it could be problematic in terms of justification or interpretability –so something would be lost if we incorporate it–.

There are to date not many studies on mobile app use. Unfortunately, the authors have missed several of them. There are some studies on the motives of use (Botnen et al., 2018; LeFebvre, 2018; Timmermans et al., 2017) and there is one study on the effects of app use on having casual sex (Grøntvedt et al., 2019) and on likelihood of getting into long-term relationships (Erevik et al., 2020). A very central finding is that sociosexual orientation (the three SOI-R components) is found to predict current use of dating apps over and above the effects of other relevant factors (Botnen et al., 2018). Where relevant, several of the above studies need to be included in the introduction part of this study. I have listed the references below. 

Response: We appreciate your recommendations. In the first version, we have tried to write a short, concrete manuscript, so certain references were not included. For this new version, the suggested references have been revised and the Introduction of the manuscript has been modified. Specifically, we have updated the references and rewritten the Introduction to include these relevant studies. This is especially clear in the paragraphs of the Introduction where we reviewed the literature about it (pp. 3-4, lines 47-83). The text is now as follows:

It is widely believed that dating apps are used exclusively for casual sex [4]. However, research on this subject suggests otherwise. In the last years, a considerable amount of research [2–8] is developed, showing that people use these apps for a wide variety of reasons, and that seeking sex is not the main one at all. The reasons given above for sex vary in different studies, including relational (e.g., friendship, love), entertainment (e.g., curiosity, boredom), and intrapersonal issues (e.g., self-validation, ease of communication).

The motivations for using dating apps are determined by the users’ individual characteristics [1]. Sociodemographic variables (i.e., sex, age, and sexual orientation) are those with a higher relationship with the use of apps [9,10]. Specifically, past literature highlighted that men [6,9], older youth [2,4,9], and members of sexual minorities [6,9,11], present higher prevalence rates for the use of dating apps.

With respect to relationship status, while some authors have found that a large proportion of people in a relationship are dating apps users [4,12,13], other studies have found that being in a relationship shows a negative and large association with current (last three months) use, but not associated with previous use [9]. Those discrepancies can be partially explained by the timeframe considered to mark participants as dating apps users. For instance, Lefebvre [4] explicitly indicated that with her data collection protocol current relationship status of the participants in may or may not reflect their status when using Tinder. Orosz et al. [13] considered as users those who had used Tinder at least once in their lifetime, so, again, current relationship status is was the same as status when using dating apps.

Personality traits is also related with the use of dating apps, but its relevance is lower [9]. A variable that seems with an important relationship with dating apps is sociosexuality, understood as an individual´s disposition to engage in sexual relations without closeness or commitment [14–18]. Some studies, like those of Botnen et al. [8] and Grøntvedt et al. [19] have found that people with unrestricted sociosexuality (i.e., short-mating orientation) report to use these apps more.

Continuing with the influence of individual differences, the literature has paid particular attention to mating preferences and orientations. Mating is a lifelong process [20,21] with great implications for future life [22,23].

Given the solidity of the measurements used for this study, and the high number of sexual minority students, the authors should be able to re-analyze and present their findings more in line with the prior studies and theoretical models with particular focus on possible differences between heterosexual and sexual minority students of both gender. However, as the paper reads now, I would not recommend it for publication in PLOS ONE. 

Response: The reviewer suggests a "particular focus on possible differences between heterosexual and sexual minority students of both gender". First, we have to note that the relationship between dating apps use and sexual orientation or gender was already analyzed in the previous version of the manuscript. See, for instance, Table 1, where the association among all those variables was reported. Second, the goal of the manuscript, from our point of view, is clear: Do young dating app users and non-users differ in mating orientations? Our focus has to be directed to those analyses required to provide an answer to our research question. We have compared, both with bivariate associations and high multiple linear regression, users and non-users in mating orientations. We do not know what more is expected.

Minor:

In the description of inclusion criteria, I would prefer a more straightforward presentation (e.g., use 'single', rather than 'not having a partner at the time of the study').

Response: Following your recommendation, we have modified this throughout the manuscript.

Relevant references:

Botnen, E. O., Bendixen, M., Grøntvedt, T. V., & Kennair, L. E. O. (2018). Individual differences in sociosexuality predict picture-based mobile dating app use. Personality and Individual Differences, 131, 67-73. doi:https://doi.org/10.1016/j.paid.2018.04.021

Erevik, E. K., Kristensen, J. H., Torsheim, T., Vedaa, Ø., & Pallesen, S. (2020). Tinder Use and Romantic Relationship Formations: A Large-Scale Longitudinal Study. Frontiers in Psychology, 11(1757). doi:10.3389/fpsyg.2020.01757

Grøntvedt, T. V., Bendixen, M., Botnen, E. O., & Kennair, L. E. O. (2019). Hook, Line and Sinker: Do Tinder Matches and Meet Ups Lead to One-Night Stands? Evolutionary Psychological Science. doi:10.1007/s40806-019-00222-z

LeFebvre, L. E. (2018). Swiping me off my feet: Explicating relationship initiation on Tinder. Journal of Social and Personal Relationships, 35(9), 1205-1229. doi:10.1177/0265407517706419

Response: Thank you very much for your suggestions. We have included these references throughout the manuscript.

 

COMMENTS BY REVIEWER #2

The topic of the article is of importance for potential as well as actual dating app users, and is of everyday interest. The article tries to distinguish between recent users and non-users regarding the attitude towards short-term mating and long-term mating of a Spanish sample of (single) students.

Response: We would like to thank the Reviewer 2 for the time and effort invested in reviewing our manuscript and for his/her comments. Your suggestions have been very useful to improve our work.

Some general aspects:

Behavior is not equal to asking if someone had the app installed during the last 3 months. Differences, like you mentioned out, might

Response: Unfortunately, this comment is incomplete, so we cannot be sure to have understood it correctly. We will assume that it goes in line with the comment by the Editor: "Reviewer 2 points out conceptual issues like treating having downloaded the app as a behavior when it is behavioroid at best (i.e., self-reported behaviors)". We agree that this is self-reported behavior. In the Methods section was fully disclosed that dating apps use was assessed via self-report. It is not uncommon to describe as 'behavior' information collected via self-reports. For instance, in this area of research, sociosexual behavior is usually collected with questionnaires like the SOI-R, with one of its dimensions labeled Behavior.

Perhaps the reviewer tried to point to the fact that 'dating app use' is not a unitary behavior as we measured it. We agree with this point. We included these lines in the Discussion section (p. 15, lines 332-336):

The study has a number of limitations. The use of dating apps was evaluated without delving into the variety of uses, from those who used it on a single afternoon as a joke among friends to those who used it for months looking for a romantic relationship. So, what we treated a unitary (self-reported) behavior –dating apps use– included, in fact, important differences in motivations or intensity.

Introduction:

49-53:

Mating is not limited to (older) youth, further Ranzini and Lutz had a age range from 16 to 40, LeFebvre from 18 to 34 years. Dating orientation is of interest (DAs/tinder) do not differ. You can look for both.

Response: Thank you very much for your comment. Of course, mating is not limited to (older) youth. We use this sentence because our study is focused on the youth. Nevertheless, in the new version, the sentence has been changed to point out that mating is a lifelong process (p. 4, lines 82-83):

Mating is a lifelong process [20,21] with great implications for future life [22,23]

55-56ff:

Mating is not limited to youth, esp. not compared to the evolutionary background mentioned by Buss. Following Buss' idea, which is hard to falsify, maybe, the ongoing concept of looking at short-term and long-term mating as not being opposite poles (as e.g. Kirkpatrick [14]) could already be mentioned in this context or refer to this statement. Further, potential argumentation based on evolutionary psychology often overlooks cultural and social component, explaining the majority of some effects (Eagly & Wood, 1999, Norenzayan & Heine, 2005). Also true for 81-84 [resp. 24 & 25]. Problematic on this is also your big part of non-heterosexuals (maybe more using for motive, mentioned by your literature, and not by evolutionary reasons).

Response: Thank you very much for your comments, but we have had problems to fully understand it. With regard to what we consider to be the raised points by the reviewer here, as for age and the evolutionary process, the previous answer explained the changes made in the manuscript to make it clear that mating is present throughout life (p. 4, lines 82-83).

“Mating is a lifelong process [20,21] with great implications for future life [22,23].”.

Regarding sexual orientation, already in the first version of the manuscript it was pointed out that (1) people from sexual minorities use dating apps and casual sex more than heterosexual people (now in p. 3, lines 53-57); (2) and that for evolutionary reasons and for reasons of non-interest in procreation, people from sexual minorities tend to have a greater short-term orientation than heterosexual people (now in p. 5, lines 109-121). 

With respect to the "ongoing concept of looking at short-term and long-term mating as not being opposite poles", we already introduced this idea in the previous version of the manuscript (now in pp. 4-5, lines 91-108):

This traditional view of mating orientation has been criticized by some authors, such as Jackson and Kirkpatrick [24], who claimed that short-term and long-term orientation do not have to be opposing poles, but dimensions that, while negatively related, can be differentiated. Thus, for example, it is possible to desire a stable relationship and, while seeking it, maintain multiple sexual relationships without commitment [27,28]. The conception of sociosexuality has also be examined. Different researchers have shown the appropriateness of abandoning the classic unidimensional stance of short-term orientations [18] and paying attention to a multidimensional perspective [15]. This includes sociosexual behavior (i.e., past sociosexual behavior), attitudes (i.e., beliefs about casual sex), and desire (i.e., arousal due to chances of casual sex). However, it is still common that researchers continue to study mating strategies like opposing poles and sociosexuality from a unidimensional approach when they analyze demographic correlates.

(98-100: Using "i.e." leaves the reader wondering why used three times in a row. Leaving out?)

Response: Following your recommendation, we have removed these “i.e.”.

94-96 & 178-181-:

"A condition for being an effective option would be that dating apps users are long-term oriented or, at least, as long-term oriented as the general population." This question can neither be addressed, nor answered in the sample, see next point.

Response: Following your recommendation we have modified this sentence (pp. 5-6, lines 128-130): “A condition for being an effective option would be that dating apps users are long-term oriented or, at least, as long-term oriented as the non-users”. However, we have not been able to find the “next point” that was relevant to this issue. Thus, we only changed this issue regarding the first part of the reviewer’s comment.

Materials & Methods

Participants & Procedure

A major point is the claiming for generalizability of the sample while there were two exclusions made: Once: Age was limited from 18 to 26. 

Response: Thank you for your comment. We are unsure about what is the problem with the final sample age. From our understanding, there are two options. First, our sample cannot be generalized to all the population. We fully agree with this. That is the reason why, already in the previous version of the manuscript, we included in the Discussion section (now in p. 15, lines 336-340):

Other limitations were related to the representativeness of the sample and the generalization of the results. Among the final participants, the sample was mostly female, aged between 18 and 26, single and from a single university, making the results difficult to generalize to all university students and, still less to young non-university students.

Second, if the problem is to exclude those outside that age range, this decision was made to be consistent and coherent with the inclusion criteria of previous studies conducted with university samples (see, for example: Barrada et al., 2018; Barrada et al., 2019; Castro and Santos-Iglesias, 2016; Castro et al., 2020; Fernández del Río et al., 2019). This appears in the description of the study's inclusion criteria (now, in p. 6, lines 144-19). We decided to remain consistent across studies to reduce researchers’ degrees of freedom and, thus, avoid potential p-hacking. The exclusion criteria concerning age are predetermined and consistent in all our studies.

Second: Only people considering themselves as men or women were included in the analyses (they also could be dropped and considered as missing in the regression).

Response: As for those who do not identify themselves as men or women, as can be seen in the section Participants (now, in p. 6, lines 144-19), there were 13 people from an initial sample of 1,996, which represents 0.65% of that sample. Due to the small size of this sample of participants (already very small before other exclusion criteria were applied), it was decided not to take it into account for the final sample. With such a limited sample size, statistical power would be very low. Also, it is not possible to combine this group with any of the two majority gender identities.

Plus: The considerations seems to be hetero- and non-heterosexual (maybe using the Kinsey Scale next time?), as roughly one out of four of the sample was considering themselves as not completely heterosexual and no other preferences were offered, this label would seem to be more fitting and also look less judging, see also 276-277, were this wording was used.

Response: We are sorry but have not been able to fully understand this comment. In lines 276-277, we stress a strong finding according to previous literature: there are differences in mating orientation according to sexual orientation. Heterosexual people tend to show a greater long-term orientation, while people from sexual minorities have a greater short-term orientation. On the other hand, as it appeared in the initial submission in the description of the sociodemographic instrument used (now, in p. 7, lines 178-180), we have asked about sexual orientation and different answers were given (heterosexual, homosexual, bisexual, other; not "not completely heterosexual"). We believe that sexual orientation has been given the relevance it deserves: its importance has been justified in the introduction, the results relating to it have been commented on in the discussion, and it has been recognized in the limitations of the study that grouping heterosexuals/sexual minorities together means a loss of wealth and information on the specific way homosexuals and bisexuals behave.

This means: The very limited category of young, studying, hetero- or non-heterosexual men and women is not easy to be generalized beyond itself and it cannot answer the questions about the long-term orientation differences of the population and not answer it in itself.

Response: Again, we have had problems to fully understand the comment. The use of the different inclusion/exclusion criteria is well-justified in the manuscript. All these aspects (except the non-recognition of men and women, which is already explained in the section Participants) are pointed out in the limitations of the study, recognizing that the characteristics of the sample and of the inclusion/exclusion criteria can affect the generalization of the results. We never tried to present our sample as representative from the general population, so we do not fully understand what is the point here.

E.g. the preference for older mates is not lacking the older part in the sample.

Response: We are sorry but we also have had problems to fully understand this comment. Thus, we do not make any change in the manuscript regarding this issue.

Next: The limited age span is rather a categorical than a metric variable and therefore overemphasizing potential effects of age.

Response: From a statistical point of view, we do not understand what the reviewer is trying to indicate here. Age, up to now, is a variable represented in a ratio scale, with nine different options, from 18, 19, 20... to 26. What is clear is that the difference between 18 and 19 years is exactly the difference from 25 to 26 years. This equality of distance between adjacent scores is not a property of categorical variables, so we cannot agree with the reviewer's concern.

People in a relationship are usually typical dating app users (Freyth & Batinic, 2021; Hobbs et al, 2017; LeFebvr, 2018; Orosz et al. 2016), regards Grindr this can be assumed to, as more relationships are considered to be open. Freyth & Batinic further could not find a difference of the relationship status regarding using and not-using dating apps, but also no difference in actual dating app using behavior.

Response: Thank you very much for your comment. We are unsure about what the reviewer expects us to modify by considering this comment.

Regarding dating apps use and relationship status:

- LeFebvre (2018). As Lefebvre indicates the way how she assessed relationship status was not indicative of relationship status when using Tinder: "participants characterized their current relationship status (may/may not reflect their status when using Tinder)" (p. 1211). 

- Freyth and Batinic (2021). It is unclear for us which time frame these authors used to consider a person dating apps user. Apparently, anyone who "logged into one or more of the three most used dating apps in Germany during the last 12 months" (p. 3), although it is unclear if this refers to only tracked participants or to all participants. So, again, the relationship status when data were collected may or may be no different from the status during the previous 12 months.

- Orosz et al. 2016. As these authors indicate "[t]he targets group of the questionnaire was people who have used Tinder at least once in their lifetime". Again, what we know is that some of the participants were, when data were collected, in a relationship, but we do not know how many were in a relationship when using Tinder or dating apps.

Please, take into consideration these results from Castro et al. (2020): For those in a relationship, the probability of having never used dating apps is 0.79; the probability of being a previous user (not in the last three months) is 0.17; and the probability of being a current user (in the last three months) is 0.04. So, mixing current and previous user in a livelong group of users can lead to a severe overestimation of the presence of people in a relationship usually typical dating app users.

Other studies have used what we considered as a better approach to evaluate the association of relationship status and dating app use. For instance:

- Timmermans & Courtois (2018). They found that 82% of their sample of current Tinder user were single. Importantly, part of the sample was collected by posting the survey link in confessions pages (p. 62), which could lead to an overestimation of Tinder users in a relationship, as single users have less to 'confess'.

- Castro et al. (2020). That study shares the limitations in terms of representativeness with the current manuscript and 'current users' used a three-months timeframe.

Considering all this, we cannot agree with the reviewer's sentence that "[p]eople in a relationship are usually typical dating app users".

We have included a paragraph in the Introduction about these relevant points (pp. 3-4, lines 58-74):

With respect to relationship status, while some authors have found that a large proportion of people in a relationship are dating apps users [4,12,13], other studies have found that being in a relationship shows a negative and large association with current (last three months) use, but not associated with previous use [9]. Those discrepancies can be partially explained by the timeframe considered to mark participants as dating apps users. For instance, Lefebvre [4] explicitly indicated that with her data collection protocol current relationship status of the participants in may or may not reflect their status when using Tinder. Orosz et al. [13] considered as users those who had used Tinder at least once in their lifetime, so, again, current relationship status is was the same as status when using dating apps.

Following this rationale, recognizing the relevance of the relationship status, in this study we only considered single participants, as justified in the section Participants (now, in pp. 6-7, lines 150-162). We have added to that paragraph, on the one hand, that not only the profiles, but also the reasons for the use of the dating apps of some and others are different and, on the other hand, the references that justify it:

We discarded the participants involved in a relationship for two reasons. First, because among people in a relationship, those who had used apps in the last three months were a very small minority (n = 33, 4.1%), so its limited sample size prevented any further analysis. Second, because we understood that, among dating apps users, the profiles and motives of using dating apps of those who were or were not in a relationship had to be very different [35,36].

Concluding: The assumptions for excluding the data seems arbitrary and partwise odd. The analysis would be easier to generalize if the sample wasn't reduced this way or theoretical reasons to do so would be offered.

Response: We honestly believe that the criteria for inclusion/exclusion from the study have been justified and that everything that could affect the representativeness of the sample and the generalization of the results has been included as limitations in the Discussion section. Thus, we do not make additional changes

regarding this comment (excepting the aforementioned in previous comments and those according to suggestions by reviewer #1). For us, it is difficult to understand that the exclusion criteria can be considered “arbitrary” when the reasons behind any decision were clearly described. The reviewer may disagree, but that is not the same as writing that our assumptions were “arbitrary”.

Measures

The question about using apps in the last 3 months is probably a too short window and giving no information about the way of using the apps.

Probably it is useful to talk about "recent users"?

Response: Could the reviewer justify why he or she considers the last three months as a too short window? From our point of view, this timeframe is an adequate compromise between two needs. We have detailed this in the Method section (pp. 7-8, lines 181-190).

We used a timeframe of three months as what we considered a compromise between two needs: To consider current users while still having a large enough sample size. With longer timeframes, the meaning of 'current use' is diluted. With a much stricter timeframe, the number of current users would not be enough for the intended analysis, while the meaning of 'current use' could be misleading (consider the case if you ask for use in the last 24 hours and a very active user without Internet connection in the previous day).

Data Analysis/Results

176-181 Users/non-users on long-term mating orientation: "considered as small effect sizes". As the CI includes zero, no further reports would be necessary. Further, this part of reporting could be headlined separately (descriptive?), before the regression is presented.

Response: Again, we are surprised by the reviewer’s comment. Could the reviewer, please, provide a reference about why no further reports are needed when the effect is statistically non-significant? It is basic statistical knowledge that non-rejecting the null hypothesis (no effect in the population) is not equivalent that affirming the null hypothesis. The confidence interval describes all the values that cannot be discarded given the available evidence. We cannot discard the 0 effect (that is why we are not rejecting the null hypothesis), but we also cannot discard other values. We cannot say that we have found evidence of no effect, but that we have found evidence of no effect or very small effect.

Importantly, no statistically significant differences in long-term orientation scores were found as a function of using or non-using dating apps and the confidence interval only included what could be considered as null or small effect sizes (d = –0.11, 95% CI [–0.27, 0.06], p = .202).

No conclusions should be drawn in the results, e.g. 184: "short-term behavior". The analysis is referring to the SOI-R, which is considered to be a short-term mating measure, yet the results should be referring to the scale.

Response: It is clearly noted that the results are referring to SOI-R (see p. 9, lines 233-234): "With respect to mating orientation, those using apps showed higher scores in all three SOI-R dimensions". If we assume an adequate validity of the SOI-R (and there is no reason to doubt about this, as several previous publications have validated this scale), we cannot find the problem of writing 'short-term behavior', mainly when it is completely clear that we are using that dimension from the SOI-R.

Also, the whole section could be shortened, as just the correlation table is reported from 168-193.

Response: For us, it was unclear what the reviewer expects us to shorten. We described in the main text the Table 1. From our understanding of what is a correct redaction of a manuscript, results should not be only presented in tables, but also adequately described in the main text (e.g., APA Guidelines). Consequently, this is what we did.

On the Regression: First, it looks odd compared to Castro (2020), that the analysis was not included in there, and/or second, that is was compared to the results. As Castro did show, differences regarding age, gender and sexual minority/heterosexuality have already be shown in the data set.

Response: The research questions that we tried to address in Castro et al. (2020) and the present manuscript are different, so the current results would not have fitted in Castro et al. (2020). It is different to try to describe the sociodemographic and personality characteristics of dating apps users (Castro et al., 2020) than to try to know if users and non-users differ in mating orientation.

Next, the SOI-R should be included in one single multilinear regression, including long-term mating orientation. As mentioned before, age should probably be controlled in the model (once excluded and then once included, but maybe as dichotomous variable by e.g. mediansplit).

Response: This potential analysis would be interesting, for sure, but not for our research question. If the reviewer considers that our analyses are not suited to respond to our research question (not to other potentially interesting research questions), we would like to see a clear description of those proposed analyses and a justification for them. Given that the reviewer has not justified this point and has just described what he/she would like to see or would have done by himself/herself, we have found no reason to do this. The goals of our manuscript are clear; not especially ambitious, but clear: to compare users and nonusers in mating orientations. The proposed analysis is not the one needed to try to provide an answer to our research question.

From a statistical point of view, it has been known for a very long time that dichotomizing variables is not a good practice (Cohen, 1983).

The SOI-R did report differences between men and women (expected). Yet in the presented regression (table 2) the results are unexpected towards Penke & Asendorpf (2008, p. 14). How come? How does this look in the complete model?

Response: Again, we have had problems to fully understand the comment. Thus, we do not perform any change regarding this issue.

Discussion:

e.g. 222-229: The discussion and also the theory are referring to the motives of users. Yet no motives are analyzed, but attitudes. The link between both is missing in the article, please provide some further literature or concentrate on attitudes.

Response: We regret not to be agreed on this point. From the beginning of the manuscript, we discuss the reasons for the use of dating apps and that, despite the existing stereotype, they are used for a wide variety of reasons, not only for the search for casual sex. Given this starting situation, we analyze whether there are differences in mating orientations (short-term, long-term) depending on whether or not you are a user of dating apps. Our approach is according to previous research performed on this issue and, until the best of our knowledge, there is no alternative approach regarding this issue that has demonstrated to be better than the present one. Thus, we do not perform changes in the manuscript according with to this comment.

Severe: The reference of single university students is missing before (title or at least in the first half of the abstract).

Response: Thank you very much for this comment. Following your recommendation, we have included this issue in the first half of the abstract (p. 2, line 28): “Participants were 902 single students from a mid-size Spanish university…”. Besides that, in the Participants and Procedure section, we stated: “This study was part of a larger project carried out in a Spanish university that aimed to explore several aspects of the sexuality of young students” (now, in p. 6, lines 142-143). Furthermore, we said “the author´s university” (p. 7, line 170), and this issue is pointed out in study’s limitations (now, in p. 15, lines 336-340).

235: "Conclusions". As correlations are presented and hypothesis tested in a cross-sectional study, the wording might be chosen more thoughtful.

Response: We have carefully checked this section and we have not been able to find any causal claim. We have tried to describe our results as 'differences', 'correlations' or 'associations'. In case we missed any causal claim, we will change it.

And: The sample was heavily reduced compared to the presented references.

250-254: Better part of the sample section. The differences to other samples should then be mentioned in the limitations.

Response: Thank you for your recommendation. We’ve included additional information in the Limitations section, to include all the issues mentioned by the reviewers (now in pp. 15-16, lines 341-365):

The study has a number of limitations. The use of dating apps was evaluated without delving into the variety of uses, from those who used it on a single afternoon as a joke among friends to those who used it for months looking for a romantic relationship. So, what we treated a unitary (self-reported) behavior –dating apps use– included, in fact, important differences in motivations or intensity. Other limitations were related to the representativeness of the sample and the generalization of the results. Among the final participants, the sample was mostly female, aged between 18 and 26, single and from a single university, making the results difficult to generalize to all university students and, still less to young non-university students. 

Concerning to sexual orientation, two aspects should be noted. First, the high proportion of participants from sexual minorities, more than 30% of the final sample. This could be considered as a lack of representativeness of our sample. We consider that an alternative interpretation is possible. This study shares with previous studies the same sampling approach and population (Spanish university students with the same age range and from the same university). We will show the time of data collection and the proportion of sexual minority participants: November 2018, 27.0% [14], December 2017, 22.5% [10], May 2016, 14.7% [37], April 2016, 12.7% [34], October 2013, 8.6% [38]. A clear trend is found. The proportion of sexual minority participants is steadily increasing in our samples. We can imagine two options to explain this. First, our surveys are not just biased by sexual orientation (higher probability of participation for non-heterosexual people), but also that bias is growing. We cannot find any theoretically plausible explanation for this potential change of bias across time. Second, in fact in the population of university students (Spain, a single university) the presence of non-heterosexuality is increasing. This second alternative would imply that the problem of representativeness is more apparent that real. Further research is needed to clarify this point. In any case, in our regression analyses we included sexual orientation as covariate. In addition, to facilitate the analyses, we decided to group participants into heterosexuals and non-heterosexuals, thus losing the nuances related to the behavior of members of sexual minorities. 

Similarly, our study shares with other studies based on self-selected samples and self-reported measures the fact that the results may be limited by response and recall bias. Finally, like most literature on the subject, this study is cross-sectional. It would be interesting to design longitudinal investigations, to assess the development and stability/change, both in the use of dating apps and in mating orientations and their associations.

255-263: Might it be possible, as pointed out before (e.g. Ranzini) that male users misrepresent themselves in the apps and also in the questionnaire as a tactique? See Freyth & Batinic (2021).

Response: Thank you for your comment. It might be possible. It is different to misrepresent oneself in the apps or to misrepresent in a questionnaire. While there are reasons to lie while presenting yourself in the dating app in order to meet more people, what would be the goal of misrepresenting in a questionnaire? And while this tendency to misrepresentation should differ by gender? Perhaps we were not able to fully understand this comment, but we do not know what changes are required in the manuscript to satisfy this concern.

263ff: Age correlations might be arbitrary (see above).

Response: We regret again our incomprehension. Could the reviewer justify this point? That the age is range-restricted is noted in the Method section and commented in the Discussion. Age is an interval/ratio variable, so there is no reason to include it in a correlation/regression analysis. As we previously noted, dichotomizing it would be a mistake.

94-96, 290: One might wonder why the motivation and the attitude are sufficient to report successful behavior on finding long-term partners, as looking, searching and finding are different behavioral styles.

Response: As in the previous comment, we are not able to follow your rationale. We have not said that users will find long-term partners, just that users are do not have a negative attitude towards finding those kinds of partners.

If further variable are available in the data, please include in the analyses and results.

Response: The open database and code files for the analyses are available at the Open Science Framework repository (https://osf.io/phy4n/). There is no reason to use all the available variables of a data collection, but those that are relevant for the proposed research question. Commonly, a single manuscript does not exhaust all the collected information or potentially relevant research questions.

Mentioned sources:

Eagly, A. H., & Wood, W. (1999). The origins of sex differences in human behavior: Evolved dispositions versus social roles. American psychologist, 54(6), 408.

Freyth, L., & Batinic, B. (2021). How bright and dark personality traits predict dating app behavior. Personality and Individual Differences, 168, 110316. doi.org/10.1016/j.paid.2020.110316

Norenzayan, A., & Heine, S. J. (2005). Psychological universals: What are they and how can we know?. Psychological bulletin, 131(5), 763.

Response: Thank you very much for your suggestions.

References

Altmejd, A., Dreber, A., Forsell, E., Huber, J., Imai, T., Johannesson, M., ... & Camerer, C. (2019). Predicting the replicability of social science lab experiments. PloS One, 14(12), e0225826. https://doi.org/10.1371/journal.pone.0225826

Barrada, J. R., & Castro, A. (2020). Tinder users: Sociodemographic, psychological, and psychosocial characteristics. International Journal of Environmental Research and Public Health, 17, 8047. https://doi.org/10.3390/ijerph17218047

Barrada, J. R., Castro, A., Correa, A. B., & Ruiz-Gómez, P. (2018). The tridimensional structure of sociosexuality: Spanish validation of the Revised Sociosexual Orientation Inventory. Journal of Sex & Marital Therapy, 44, 149-158. https://doi.org/10.1080/0092623X.2017.1335665

Barrada, J. R., Ruiz-Gómez, P., Correa, A. B., & Castro, A. (2019). Not all Online Sexual Activities are the same. Frontiers in Psychology, 10, 339. https://doi.org/10.3389/fpsyg.2019.00339

Castro, A., Barrada, J. R., Ramos-Villagrasa, P. J., & Fernández del Río, E. (2020). Profiling dating app users: Sociodemographic and personality characteristics. International Journal of Environmental Research and Public Health, 17, 3653. https://doi.org/10.3390/ijerph17103653

Castro, A., & Santos-Iglesias, P. (2016). Sexual behavior and sexual risks among Spanish university students: a descriptive study of gender and sexual orientation. Sexuality Research and Social Policy, 13, 84-94. https://doi.org/10.1007/s13178-015-0210-0

Cohen, J. (1983). The cost of dichotomization. Applied Psychological Measurement, 7, 249–253. https://doi.org/10.1177/014662168300700301

Fernández del Río, E., Ramos-Villagrasa, P. J., Castro, A., & Barrada, J. R. (2019). Sociosexuality and bright and dark personality: The prediction of behavior, attitude, and desire to engage in casual sex. International Journal of Environmental Research and Public Health, 16, 2731. https://doi.org/10.3390/ijerph16152731

Freyth, L., & Batinic, B. (2021). How bright and dark personality traits predict dating app behavior. Personality and Individual Differences, 168, 110316. https://doi.org/10.1016/j.paid.2020.110316

LeFebvre, L. E. (2018). Swiping me off my feet: Explicating relationship initiation on Tinder. Journal of Social and Personal Relationships, 35, 1205-1229. https://doi.org/10.1177/0265407517706419

Open Science Collaboration. (2015). Estimating the reproducibility of psychological science. Science, 349(6251).

Orosz, G., Tóth-Király, I., Böthe, B., & Melher, D. (2016). Too many swipes for today: The development of the Problematic Tinder Use Scale (PTUS). Journal of Behavioral Addictions, 5, 518-523. https://doi.org/10.1556/2006.5.2016.016

Timmermans, E., & Courtois, C. (2018). From swiping to casual sex and/or committed relationships: Exploring the experiences of Tinder users. Information & Society, 34, 59-70. https://doi.org/10.1080/01972243.2017.1414093

---

## [Decision Letter · Decision Letter 1]

31 Dec 2020

PONE-D-20-29867R1

Do young dating app users and non-users differ in mating orientations?

PLOS ONE

Dear Dr. Castro,

Thank you for submitting your manuscript to PLOS ONE. After careful consideration, we feel that it has merit but does not fully meet PLOS ONE’s publication criteria as it currently stands. Therefore, we invite you to submit a revised version of the manuscript that addresses the points raised during the review process. I was unable to secure the same reviewers but the third reviewer felt favorably towards your paper. She thought you did a reasonable job addressing comments but asks for several more changes.

We look forward to receiving your revised manuscript.

Kind regards,

Peter Karl Jonason

Academic Editor

PLOS ONE

Reviewers' comments:

Reviewer's Responses to Questions

**Comments to the Author**

1. If the authors have adequately addressed your comments raised in a previous round of review and you feel that this manuscript is now acceptable for publication, you may indicate that here to bypass the “Comments to the Author” section, enter your conflict of interest statement in the “Confidential to Editor” section, and submit your "Accept" recommendation.

Reviewer #3: (No Response)

2. Is the manuscript technically sound, and do the data support the conclusions?

Reviewer #3: Yes

3. Has the statistical analysis been performed appropriately and rigorously? 

Reviewer #3: Yes

4. Have the authors made all data underlying the findings in their manuscript fully available?

Reviewer #3: Yes

5. Is the manuscript presented in an intelligible fashion and written in standard English?

Reviewer #3: Yes

6. Review Comments to the Author

Reviewer #3: This is an interesting study conducted on a large sample assessing dating app use and mating orientations. Of particular interest is that the authors find no correlation between dating app use and LTMO. The authors have demonstrated a good attempt to address most of the reviewers feedback. However, I do have some recommendations for minor revisions.

The analysis is appropriate to answer the research question - however, what was the rationale for not including STMO? Inclusion of this measure would have provided additional, interesting information (e.g., as the author themselves notes, that STMO and LTMO do not exist independently).

Although I appreciate sociosexual might share variance with STMO, my understanding is: Restricted sociosexuality (i.e., preference for sex within long-term and committed relationships) and unrestricted sociosexuality (i.e., preference for short-term and no-strings-attached sex). Thus, we could argue that sociosexuality will also share variance with LTMO. Given the low(ish) correlations between sociosexuality and LTMO in Table 1, clearly they share variance but are still distinct. Thus, STMO could also have been included in addition to SOI-R.

The inclusion of a measure of STMO could have added richness to results. Apps used and SOI-B behaviour have a particular high correlation (Table 1). It is interesting that LTMO is not correlated to app use, but they are sociosexually unrestricted in their behaviour. All other SOI scales are correlated quite highly too. Theoretical implications of this suggest to me that perhaps it is time researchers step away from the conceptualisation of unrestricted = STMO, restricted = LTMO. People might be looking for a long-term partner, but also have an unrestricted sociosexuality.

In sum, if the authors did not include the STMO in a larger data set and it cannot be included, I think a discussion of why SOI is included and not STMO is required.

It is not a problem to have sampled young adults (or adopting Arnett categorisation, emerging adults); however, the rationale for this sample needs to be stronger. The choice for this age range needs to be embedded in the introduction discussion of orientations. Why, in particular, are you interested in young adults? Does their app use appear to be different? Their mating orientations? Given the evolutionary perspective applied, it could be particularly important to provide a rationale for assessing orientations of emerging adults (e.g., fertility?)

Finally, the authors have adequately addressed reviewer concerns about generalisability. The authors include good discussion, particularly in relation to the increasing % of sexual minority participants. However, although this trend is applicable in Spain, the authors have not really addressed if this is generalised to other countries? I also do not understand the statement (line 335): 'the problem of representativeness is more apparent that real'.

7. PLOS authors have the option to publish the peer review history of their article (what does this mean?). If published, this will include your full peer review and any attached files.

Reviewer #3: No

---

## [Author Response · Author response to Decision Letter 1]

15 Jan 2021

PONE-D-20-29867R1

Do young dating app users and non-users differ in mating orientations?

PLOS ONE

First of all, we would like to thank the editor and the reviewer 3 for the time and effort invested in reviewing this manuscript. Your comments have been very useful to improve our work. 

COMMENTS BY REVIEWER #3

This is an interesting study conducted on a large sample assessing dating app use and mating orientations. Of particular interest is that the authors find no correlation between dating app use and LTMO. The authors have demonstrated a good attempt to address most of the reviewers feedback. However, I do have some recommendations for minor revisions.

Response: We would like to thank Reviewer 3 for the time and effort invested in reviewing our manuscript and for her comments. Your suggestions have been very useful to improve our work. 

The analysis is appropriate to answer the research question - however, what was the rationale for not including STMO? Inclusion of this measure would have provided additional, interesting information (e.g., as the author themselves notes, that STMO and LTMO do not exist independently).

Response: In this point we are unsure about what Reviewer #3 meant by STMO. We will distinguish between two potential meanings:

1) STMO as the subscale from the questionnaire developed by Jackson and Kirkpatrick (2007).

Among the reasons why Jackson and Kirkpatrick (2007) developed their questionnaire we can highlight two. First, because they wanted to clearly separate between short-term and long-term mating, as previous literature has tended to consider both constructs as opposite extremes from the same dimension. They considered and showed that it was not the case. Second, because the original SOI questionnaire (Simpson & Gangestad, 1991) mixed sociosexual attitudes (attitudes in favor of casual sex) and sociosexual behaviors (number of previous sexual partners). They proposed STMO items, which included three items from the SOI– tapped attitudes/comfort with casual sex and avoided the multidimensionality of the SOI, but it did not assess behaviors.

Importantly, Penke and Asendorpf (2008) presented, shortly after Jackson and Kirkpatrick's (2007) article was published, an improved SOI version, the SOI-R. In this improved version, the problems of multidimensionality of the SOI are also considered, but, instead of reducing the short-term orientation to attitudes, the SOI-R dimensions of behavior and desire were also included. So, the SOI-R questionnaire –the one used in this study– covers the same as the STMO plus two additional aspects of short-term orientation. Previous research has shown the advantages of considering these three interrelated but different aspects (e.g., Barrada et al., 2018). As all the SOI-R dimensions have been shown to present adequate psychometric properties and its content is wider than both the original SOI and the STMO, we found no reason to include STMO.

2) STMO as the construct of short-term mating orientation.

We included a measure of short-term mating orientation: the SOI-R. Although we acknowledge that the terms of 'sociosexuality' and 'short-term orientation' have not been always clearly defined, we consider that there are several reasons to understand unrestricted sociosexuality as equivalent to short-term orientation (e.g., Buss & Schmitt, 2019; Penke & Asendorpf, 2008). In the Introduction section we clearly noted that we would use both terms as interchangeable:

Some studies, like those of Botnen et al. [8] and Grøntvedt et al. [19] have found that people with unrestricted sociosexuality (i.e., short-mating orientation) report to use these apps more.

Also, in the Discussion we had written:

Thirdly, among the contributions of the article should be highlighted the assessment of sociosexuality from a multidimensional point of view, distinguishing between behavior, attitudes, and desire, following the recommendations of other authors [15,38]. It has been shown that the three dimensions of the construct, understood as short-term orientation,

The SOI-R measures sociosexuality, that is, it measures short-term orientation. As indicative of this, as we have said, the STMO questionnaire and the SOI share several items. One of the advantages of the SOI-R is that it clearly differentiates behavior, attitudes, and desire. Some of the research about short-term orientation have mixed these different dimensions.

In the previous version of the manuscript we discussed the differences between short- and long-term mating orientations in this way:

This traditional view of mating orientation [short-term versus long-term] has been criticized by some authors, such as Jackson and Kirkpatrick [24], who claimed that short-term and long-term orientation are not two opposing poles in a single dimension, but two dimensions that, while negatively related, can be and should be differentiated. Thus, for example, it is possible to desire or be involved into a stable relationship and maintain multiple sexual relationships without commitment [27,28]. It is also possible to have no interest in any kind of relationship.

The conception of sociosexuality has also be refined. Different researchers have shown the appropriateness of abandoning the classic unidimensional stance of short-term orientations [18] and paying attention to a multidimensional perspective [15]. This more fine-grained approach includes sociosexual behavior (i.e., past sociosexual behavior), attitudes (i.e., positive appraisal about casual sex), and desire (i.e., sexual arousal with people with whom no committed romantic relationship exists).

We have included a new paragraph to clarify some possible ambiguity in the terminology (p. 5, lines 96-108):

However, it is still common that researchers continue to study mating strategies like opposing poles and sociosexuality from a unidimensional approach when they analyze demographic and psychological correlates. There is still some theoretical confusion in the use of some terms. For instance, Penke [29] defined restricted sociosexuality as the "tendency to have sex exclusively in emotionally close and committed relationships" and unrestricted sociosexuality as the "tendency for sexual relationships with low commitment and investment" (p. 622). This conceptualization assumes that (a) restricted and unrestricted sociosexuality define a single dimension and (b) that restricted is equivalent to long-term mating orientation and unrestricted to short-term orientation. While we agree with the first assumption, we have justified that short- and long-term mating orientation are not the two extremes of a single dimension. While unrestricted sociosexuality can be understood as interchangeable with short-term orientation, restricted sociosexuality is not long-term, but lack of short-term orientation.

Although I appreciate sociosexual might share variance with STMO, my understanding is: Restricted sociosexuality (i.e., preference for sex within long-term and committed relationships) and unrestricted sociosexuality (i.e., preference for short-term and no-strings-attached sex). Thus, we could argue that sociosexuality will also share variance with LTMO. Given the low(ish) correlations between sociosexuality and LTMO in Table 1, clearly they share variance but are still distinct. Thus, STMO could also have been included in addition to SOI-R.

Response: As we have mentioned, from our point of view, it is not that sociosexual orientation shares variance with STMO, but that both are equivalent. As we have noted, both of their prototypical measures share several items. One of the reasons to use the SOI-R is that it clearly differentiates between several aspects of short-term (sociosexual) orientation, which implies that this measure offers richer information.

We have to disagree with the way how the Reviewer has characterized restricted sociosexuality ("preference for sex within a long-term and committed relationship"). By understanding sociosexuality in this way we would be considering mating orientations with a unidimensional approach, where long-term and short-term are opposite poles. As Jackson and Kirkpatrick (2007) argued and showed, and as our own results indicate, both orientations are negatively correlated, but cannot be considered as lying into a single dimension. If that was the case, our research could not offer anything new, as it has already repeatedly found that dating apps users show higher short-term orientation (e.g., Barrada & Castro, 2020). So restricted sociosexuality would be better understood as a lack of preference for short-term sex.

We agree with the idea that sociosexuality and LTMO share variance but are still distinct. We developed this idea in the Introduction.

As we have argued previously, the STMO questionnaire is partially based on the SOI (and the SOI-R on the SOI), so we find no reason to include the STMO. Also, we consider both constructs, short-term orientation and sociosexuality, as equivalent.

The inclusion of a measure of STMO could have added richness to results. Apps used and SOI-B behaviour have a particular high correlation (Table 1). It is interesting that LTMO is not correlated to app use, but they are sociosexually unrestricted in their behaviour. All other SOI scales are correlated quite highly too. Theoretical implications of this suggest to me that perhaps it is time researchers step away from the conceptualisation of unrestricted = STMO, restricted = LTMO. People might be looking for a long-term partner, but also have an unrestricted sociosexuality.

Response: As we have said, as short-term mating orientation and sociosexual orientation are interchangeable terms, we consider that nothing could be gained by including a (second) STMO measure, a redundant measure.

We agree with the idea that the conceptualization of unrestricted = STMO and restricted = LTMO is, although intuitive, wrong. That was the theoretical contribution from Jackson and Kirkpatrick (2007), a main source for our manuscript. It is not difficult to imagine people with no interest in neither short-term nor long-term or with interest in both of them.

We have included this in the Discussion section (p. 15, lines 298-302):

This points to the need to step away from the conceptualization of unrestricted sociosexuality as equal to short-term mating orientation and restricted sociosexuality as equal to long-term mating orientation [29]. As we previously noted, restricted sociosexuality is better understood as lack of short-term orientation, what is not equivalent to long-term orientation.

In sum, if the authors did not include the STMO in a larger data set and it cannot be included, I think a discussion of why SOI is included and not STMO is required.

Response: As we have said, we have theoretical reasons to consider unrestricted sociosexuality and short-term mating orientation as equivalent. For that reason, we find no reason to discuss why short-term orientation was not included as, in fact, it was.

It is not a problem to have sampled young adults (or adopting Arnett categorisation, emerging adults); however, the rationale for this sample needs to be stronger. The choice for this age range needs to be embedded in the introduction discussion of orientations. Why, in particular, are you interested in young adults? Does their app use appear to be different? Their mating orientations? Given the evolutionary perspective applied, it could be particularly important to provide a rationale for assessing orientations of emerging adults (e.g., fertility?).

Response: Thank you very much for giving us the opportunity to clarify this point. Our interest in this population of emerging adults is threefold. First of all, due to methodological reasons. As it appears in the previous version of the manuscript and we explain to one of the previous reviewers, we focus on this population (university students aged 18 to 26) to be consistent and coherent with the inclusion criteria of previous studies conducted in our research group (see, for example, Barrada et al., 2018; Barrada et al., 2019; Castro and Santos-Iglesias, 2016; Castro et al., 2020; Fernández del Río et al., 2019). This appears in the description of the study's inclusion criteria.

Second, as is highlighted in some reviews on the use of dating apps (e.g., Castro & Barrada, 2020), emerging adults are the most studied group and in which higher rates of use of these applications have been found. This idea already appeared in the previous version of the manuscript. However, following your recommendations, we have added an explanation of the differences in use and motivations for use, to highlight the relevance of focusing on this population (p. 3, lines 49-55):

The motivations for using dating apps are determined by the users’ individual characteristics [1]. Sociodemographic variables (i.e., sex, age, and sexual orientation) are those with a higher relationship with the use of apps [9,10]. Specifically, past literature highlighted that men [6,10], and members of sexual minorities [6,10,11], present higher prevalence rates for the use of dating apps. Based on age, the most studied group and in which higher rates of app use is older youth, who tend to show a wide variety of motives to use it, seeking both entertainment and casual sex or romantic partner [2,4,10].

 Finally, from an evolutionary perspective, we understand the relevance of this population for mating. We start from the assumption that mating is a lifelong process with great implications for future life, as we already pointed out in the manuscript. However, in the study of mating, special attention has been paid to some specific stages of life, such as youth. In this stage, they are made decisions (e.g., looking for a partner with whom to have a committed romantic relationship and have children, or on the contrary to seek only casual sexual relationships, or to seek both types of relationships, or not to seek neither) that can influence the rest of people's lives and determine their future relationships (that is, partner, parenthood/motherhood) and behaviors (Buss & Schmitt, 2019). Following your recommendations, we have completed the paragraph in the Introduction that talks about the importance of mating (now, on p. 4, lines 71-83).

Continuing with the influence of individual differences, the literature has paid particular attention to mating preferences and orientations. Mating is a lifelong process [20,21] with great implications for future life [22,23]. Traditionally, its relevance has been emphasized during emerging adulthood, when decisions are often made about relationships and offspring, events that have a considerable impact on peoples’ lives. [20,21]. Mating orientation, the individuals’ stated interest in committed relationships and/or in brief or uncommitted sexual relationships [24], has usually been measured through a single dimension with two opposite poles: short-term versus long-term [21]. Short-mating orientation is characterized by the search for casual sexual partners and relationships of low emotional commitment [21,24,25], and traditionally has been identified with unrestricted sociosexuality. Long-term mating orientation, on the other hand, is characterized by the desire for romantic relationships of commitment, with a strong emotional investment in the relationship and, generally, with sexual exclusivity [26].

Finally, the authors have adequately addressed reviewer concerns about generalisability. The authors include good discussion, particularly in relation to the increasing % of sexual minority participants. However, although this trend is applicable in Spain, the authors have not really addressed if this is generalised to other countries? 

Response: Thank you very much for your words. In the previous version of the manuscript, and in response to one of the reviewers, we tried to explain this increase in the prevalence of members of sexual minorities, and we indicated that our analysis included sexual orientation as covariate:

First, our surveys are not just biased by sexual orientation (higher probability of participation for non-heterosexual people), but also that bias is growing. We cannot find any theoretically plausible explanation for this potential change of bias across time. Second, in fact in the population of university students (Spain, a single university) the presence of non-heterosexuality is increasing. This second alternative would imply that the large number of non-heterosexual participants is not a problem of representativeness of the samples. Further research is needed to clarify this point. In any case, in our regression analyses we included sexual orientation as covariate. In addition, to facilitate the analyses, we decided to group participants into heterosexuals and non-heterosexuals, thus losing the nuances related to the behavior of members of sexual minorities.

This prevalence of sexual minorities is higher than that found in similar studies in other countries, both European (for example, Sumter & Vandenbosch (2019), in the Netherlands, had an 83.5% of heterosexuals and 16.5% of non-heterosexuals) as in the United States (for example, Alexopoulos et al. (2020), found 80.5% of heterosexuals and 19.5% of non-heterosexuals). Nevertheless, this prevalence is quite similar to research in other countries, like this by Tsoukas and March (2018) among Australians aged 18 to 69 (71% heterosexuals, 29% non-heterosexuals).

 Besides that, one of the latest studies published on this topic is that of Rahman et al. (2020), who assessed the prevalence of women's and men's sexual orientation in 28 nations using data from 191,088 participants. These authors found that, in terms of sexual attraction (one of de components of sexual orientation), 82.6% of men and 66.2% of women were predominantly not attracted to the same sex, while 17.4% of men and 33.8% of women were moderately or predominantly attracted to the same sex. Considering data by country, with data grouped together for men and women, it can be seen that the proportion for Spain is 73% vs. 27%, similar to that of this study and very similar to that of other countries, such as Australia (74%-26%), Finland (71%-29%), Netherlands (70%-30%), or United States (73%-27%). Therefore, it seems that in terms of sexual attraction, the data from other countries are similar to ours. As for the trend, it seems that in recent years and especially in countries with more tolerant laws towards sexual minorities, such as Spain, there is an increase in the prevalence of people who claim to belong to them, a process parallel to the decrease of stigma and an improvement in the quality of life of these people (Pachankis et al., 2016).

 Taking all of these into account, we have modified the Discussion section, briefly pointing out that the results obtained and the trend in our data may be similar to what is observed in other countries (now, in p. 16-17, lines 334-361):

Concerning to sexual orientation, two aspects should be noted. First, the high proportion of participants from sexual minorities, more than 30% of the final sample. This could be considered as a lack of representativeness of our sample. We consider that an alternative interpretation is possible. This study shares with previous studies the same sampling approach and population (Spanish university students with the same age range and from the same university). We will show the time of data collection and the proportion of sexual minority participants: November 2018, 27.0% [14], December 2017, 22.5% [9], May 2016, 14.7% [38], April 2016, 12.7% [35], October 2013, 8.6% [39]. A clear trend is found. The proportion of sexual minority participants is steadily increasing in our samples. 

We can imagine two options to explain this. First, our surveys are not just biased by sexual orientation (higher probability of participation for non-heterosexual people), but also that bias is growing. We cannot find any theoretically plausible explanation for this potential change of bias across time. Second, in fact in the population of university students (Spain, a single university) the presence of non-heterosexuality is increasing. This second alternative would imply that the large number of non-heterosexual participants is not a problem of representativeness of the samples.

This hypothesis may be supported by data on the prevalence of persons from sexual minorities found in other studies, which can be exemplified in that of Rahman et al. [40], who assessed the prevalence of women's and men's sexual orientation in 28 nations and found similar proportions to those of the present study, both in Spain (73% vs. 27%) and in other countries. There seems to be a trend toward greater self-identification as a member of sexual minorities, paralleling the decrease in stigma and the improvement in the quality of life of these people, especially in countries with more tolerant laws, as is the case in Spain [41]. However, further research is needed to clarify this point. And, in any case, in our regression analyses, we included sexual orientation as a covariate. In addition, to facilitate the analyses, we decided to group participants into heterosexuals and non-heterosexuals, thus losing the nuances related to the behavior of members of sexual minorities. 

I also do not understand the statement (line 335): 'the problem of representativeness is more apparent that real'.

Response: We have changed this sentence to (p. 17, lines 347-349):

This second alternative would imply that the large number of non-heterosexual participants is not a problem of representativeness of the samples

References

Alexopoulos, C., Timmermans, E., & McNallie, J. (2020). Swiping more, committing less: Unraveling the links among dating app use, dating app success, and intention to commit infidelity. Computers & Human Behavior, 102, 172-180. https://doi.org/10.1016/j.chb.2019.08.009

Barrada, J. R., & Castro, A. (2020). Tinder users: Sociodemographic, psychological, and psychosexual characteristics. International Journal of Environmental Research and Public Health, 17(21), 8047. https://doi.org/10.3390/ijerph17218047

Barrada, J. R., Castro, A., Correa, A. B., & Ruiz-Gómez, P. (2018). The tridimensional structure of sociosexuality: Spanish validation of the Revised Sociosexual Orientation Inventory. Journal of Sex & Marital Therapy, 44, 149–158. https://doi.org/10.1080/0092623X.2017.1335665

Barrada, J. R., Ruiz-Gómez, P., Correa, A. B., & Castro, A. (2019). Not all Online Sexual Activities are the same. Frontiers in Psychology, 10, 339. https:/doi.org/10.3389/fpsyg.2019.00339

Buss, D. M., & Schmitt, D. P. (2019). Mate preferences and their behavioral manifestations. Annual Review of Psychology, 70, 77–110. https://doi.org/10.1146/annurev-psych-010418-103408

Castro, A., & Barrada, J. R. (2020). Dating apps and their sociodemographic and psychosocial correlates: A systematic review. International Journal of Environmental Research and Public Health, 17, 6500. https://doi.org/10.3390/ijerph17186500

Castro, A., Barrada, J. R., Ramos-Villagrasa, P. J., & Fernández del Río, E. (2020). Profiling dating app users: Sociodemographic and personality characteristics. International Journal of Environmental Research and Public Health, 17, 3652. https://doi.org/10.3390/ijerph17103653

Castro, A., & Santos-Iglesias, P. (2016). Sexual behavior and sexual risks among Spanish university students: A descriptive study of gender and sexual orientation. Sexuality Research and Social Policy, 13, 84-94. https://doi.org/10.1007/s13178-015-0210-0

Fernández del Río, E., Ramos-Villagrasa, P. J., Castro, A., & Barrada, J. R. (2019). Sociosexuality and bright and dark personality: The prediction of behavior, attitude, and desire to engage in casual sex. International Journal of Environmental Research and Public Health, 16, 2731. https://doi.org/10.3390/ijerph17186500

Jackson, J. J., & Kirkpatrick, L. A. (2007). The structure and measurement of human mating strategies: Toward a multidimensional model of sociosexuality. Evolution and Human Behavior, 28, 382–391. https://doi.org/10.1016/j.evolhumbehav.2007.04.005

Pachankis, J. E., Haztenbuehler, M. L., Mirandola, M., Weatherburn, P., Berg, R. C., Marcus, U., et al. (2016). The geography of sexual orientation: Structural stigma and sexual attraction, behavior, and identity among men who have sex with men across 38 European countries. Archives of Sexual Behavior, 46, 1491-1502. https://doi.org/10.1007/s10508-016-0819-y

Penke, L., & Asendorpf, J. B. (2008). Beyond global sociosexual orientations: A more differentiated look at sociosexuality and its effects on courtship and romantic relationships. Journal of Personality and Social Psychology, 95(5), 1113–1135. https://doi.org/10.1037/0022-3514.95.5.1113

Rahman, Q., Xu, Y., Lippa, R. A., & Vasey, P. L. (2020). Prevalence of sexual orientation across 28 nations and its association with gender equality, economic development, and individualism. Archives of Sexual Behavior, 49, 595-606. https://doi.org/10.1007/s10508-019-01590-0

Simpson, J. A., & Gangestad, S. W. (1991). Individual differences in sociosexuality: Evidence for convergent and discriminant validity. Journal of Personality and Social Psychology, 60(6), 870–883. https://doi.org/10.1037/0022-3514.60.6.870

Sumter, S. R., & Vandenbosch, L. (2019). Dating gona mobile: Demographic and personality-based correlates of using smartphone-based dating applications among emerging adults. New Media & Society, 21, 655-673. https://doi.org/10.1177/1461444818804773

Tsoukas, A., & March, E. (2018). Predicting short- and long-term mating orientations: The role of sex and the Dark Tetrad. Journal of Sex Research, 55, 1206-1218. https://doi.org/10.1080/00224499.2017.1420750

---

## [Editor Report · Decision Letter 2]

18 Jan 2021

Do young dating app users and non-users differ in mating orientations?

PONE-D-20-29867R2

Dear Dr. Castro,

We’re pleased to inform you that your manuscript has been judged scientifically suitable for publication and will be formally accepted for publication once it meets all outstanding technical requirements.

Kind regards,

Peter Karl Jonason

Academic Editor

PLOS ONE
---

## [Editor Report · Acceptance letter]

22 Jan 2021

PONE-D-20-29867R2 

Do young dating app users and non-users differ in mating orientations? 

Dear Dr. Castro:

I'm pleased to inform you that your manuscript has been deemed suitable for publication in PLOS ONE. Congratulations! Your manuscript is now with our production department. 

Kind regards, 

on behalf of

Dr. Peter Karl Jonason 

Academic Editor

PLOS ONE